# User-Creator Feature Polarization in Recommender Systems with Dual Influence

**Tao Lin**[*]
Harvard University
tlin@g.harvard.edu

**Kun Jin**[*]
Google
kunjin@google.com

**Andrew Estornell**
ByteDance
andrew.estornell@bytedance.com

**Xiaoying Zhang**
ByteDance
zhangxiaoying.xy@bytedance.com

**Yiling Chen**
Harvard University
yiling@seas.harvard.edu

**Yang Liu**
University of California, Santa Cruz
yangliu@ucsc.edu

## Abstract

Recommender systems serve the dual purpose of presenting relevant content to users and helping content creators reach their target audience. The dual nature of these systems naturally influences both users and creators: users' preferences are affected by the items they are recommended, while creators may be incentivized to alter their content to attract more users. We define a model, called user-creator feature dynamics, to capture the dual influence of recommender systems. We prove that a recommender system with dual influence is guaranteed to polarize, causing diversity loss in the system. We then investigate, both theoretically and empirically, approaches for mitigating polarization and promoting diversity in recommender systems. Unexpectedly, we find that common diversity-promoting approaches do not work in the presence of dual influence, while relevancy-optimizing methods like top-$k$ truncation can prevent polarization and improve diversity of the system.

## 1 Introduction

From restaurant selection, video watching, to apartment renting, recommender systems play a pivotal role across a plethora of real-world domains. These systems match users with content they like, and help creators (those producing the content) identify their target audiences. Nevertheless, behind such success, concerns have emerged regarding possible harmful outcomes of recommender systems, in particular, *filter bubbles* [32, 5] and *polarization* [36] – outcomes with insufficient *recommendation diversity* and *creation diversity*. Recommendation diversity, meaning the diversity of the contents recommended to a user, is key to users' engagement and retention on the platform. Meanwhile, creation diversity, meaning the variety of content created on the platform, is a determinant of the platform's long-term health. In extreme cases, insufficient creation diversity can lead to consensus or polarization, where the latter can cause conflict and hatred, diminish people's mutual understanding, and cause societal crises. Therefore, from both business and social responsibility perspectives, championing and improving diversity in recommender systems is equally important as optimizing recommendation relevancy.

---

[*]The first two authors contributed equally to this work. And this work was done when Kun was at TikTok.

38th Conference on Neural Information Processing Systems (NeurIPS 2024).

There is increasing emphasis in academia and industry on investigating and improving the diversity of recommender systems, combating filter bubbles and polarization. Popular diversity-boosting approaches include applying post-processing procedures such as re-ranking [11, 47] and setting diversity-aware objectives in addition to relevance maximization [38, 44, 22, 39, 12]. These methods aim to increase the recommendation diversity for users. Assuming that the contents on the platform are static, these methods have been shown to bring diversity gain to the system.

However, an important aspect that is overlooked in the aforementioned approaches is that: users and contents on a recommendation platform are not static entities – they can be *influenced* by the recommendation made by the system. In content creation platforms like YouTube, TikTok, and Twitter, recommendations naturally affect both content users and content creators. It is well known that the exposure to recommended items can shift a user's preference [24, 26, 14]. On the other hand, the creators have the incentive to change their creation styles constantly to attract their audience better (and to make more profits from the platform) [15, 20, 23]. While the effects of recommendation on either users or creators have been investigated separately, to our knowledge no previous work considers both effects. The dual influence of recommendation on users and creators causes complicated dynamics where users and creators interact and their preferences evolve together. Such evolution might exacerbate filter bubble and polarization effects. Whether the aforementioned diversity-boosting approaches still work in a dynamic environment with dual influence is questionable.

**Our contributions**    The first contribution of our work is to define a novel, natural dynamics model that captures the dual influence of a recommender system on users and creators, which we call user-creator feature dynamics (Section 2). We leverage the users' and items'/creators' embedding vectors to represent their preferences and creation styles, and use cosine similarity to characterize the relevance of creations and users' interests (which is common in the recommender system literature and practice). This model allows us to formally reason about the impact of various design choices on the long-term diversity of a recommender system with dual influence.

Our second contribution is to demonstrate that, under realistic conditions, the user-creator feature dynamics of any recommender system with dual influence must unavoidably converge to polarization (Section 3), i.e., the preferences of users and the contents of creators will become tightly clustered into two opposite groups, significantly reducing the diversity of the system. We demonstrate that this phenomenon still occurs even after applying diversity-boosting interventions to the system.

Then, (in Section 4) we investigate some real-world designs of recommendation algorithms in order to look for techniques that mitigate polarization. Interestingly, we find that some common efficiency-improving methods, such as top-$k$ truncation, can both prevent the system from polarization and improve the creation diversity. We also provide empirical results (Section 5) on both synthetic and real-world (MovieLens) data. As predicted by our theory, we find that systems with dual influence more easily converge to polarization under diversity-boosting designs, while efficiency-oriented and relevance-optimizing designs can in fact improve the long-term diversity of the system. This could explain why polarization does not always happen in reality. Section 6 concludes.

## 1.1  Related Work

**Diversity in recommendations**    Diversity, filter bubbles, and polarization in recommendations have been important research topics in recent years, and they are closely related concepts with different focuses. On the one hand, filter bubbles are frequently defined as decreasing recommendation diversity over time [5], which describes both the process and the outcome of insufficiently diverse recommendations. On the other hand, polarization describes the negative outcome of insufficient mutual understanding between people [36]. In content platforms, an example of polarization is people creating content with strong agreement or disagreement with other content under the same topic, e.g., political opinions. To combat these negative outcomes, previous works propose diversity-boosting approaches including re-ranking [11, 47] and diversity-aware objective optimization [38, 44, 22, 39, 12, 45]. Despite having positive effects in situations where user preferences and creation styles are fixed, these approaches overlooked the dynamic nature of recommender systems and our work shows that certain approaches will make long-term outcomes worse under the dual influence.

**Opinion dynamics**    Opinion dynamics study the effect of people exchanging opinions with others on social networks [37, 17, 29, 4]. Our model of a recommender system with dual influence on users

and creators resembles a bipartite social network, and our conclusion that the system converges to polarization is conceptually similar to people reaching consensus on social networks [1, 10, 31, 46]. However, the technique we use to prove our conclusion (absorbing Markov chain) significantly differs from the main technique (stability of ODE) in the mentioned works.

**Performative effects of recommender systems**   The phenomenon that predictive systems like recommender systems can impact the individuals interacting with those systems (e.g., users and creators) is related to the literature of performative prediction [34, 18]. These impacts can be direct, such as individuals ostensibly modifying their features in order to obtain more desirable outcomes [27]. Prior works on the performative effects of recommender systems (e.g., [7, 24, 14, 41, 15, 42, 35, 20, 3, 43, 2, 23]) only consider one-sided impact, either on users or on creators. Differing from them, our work studies two-sided impacts, i.e., on both users and creators. We provide a table to compare our work with previous works in Appendix A.

## 2   Model: User-Creator Feature Dynamics

We define a *dynamics* model for user preferences and content/creator features in a recommender system. Let $U^t = [u^t_j]^m_{j=1} = [u^t_1, \ldots, u^t_m] \in \mathbb{R}^{d \times m}$ be a population of $m$ users and $V^t = [v^t_i]^n_{i=1} = [v^t_1, \ldots, v^t_n] \in \mathbb{R}^{d \times n}$ be a population of $n$ creators at time $t$, where each vector $u^t_j, v^t_i \in \mathbb{S}^{d-1}$ represent the preference/feature vector of each user and creator respectively, assumed to be on the unit sphere $\mathbb{S}^{d-1}$ with $\ell_2$-norm. Then $(U^t, V^t)$ denotes the state of the dynamics at time $t$. The dynamics evolve as follows at each time step $t \geq 0$:

**1) Recommendation:** Each user $j \in [m]$ is recommended a creator, where creator $i \in [n]$ is chosen with a probability

$$p^t_{ij} = p^t_{ij}(U^t, V^t). \tag{1}$$

While we allow a wide array of different functions $p^t_{ij}(\cdot)$, a common example of such functions is the so-called *softmax function*:

$$p^t_{ij} = \text{softmax}(u^t_j, V^t; \beta) = \frac{\exp(\beta \langle u^t_j, v^t_i \rangle)}{\sum_{i=1}^n \exp(\beta \langle u^t_j, v^t_i \rangle)}. \tag{2}$$

A larger $\beta$ means that the recommendation is more sensitive to the *relevance* of a creator to a user, measured by $\langle u^t_j, v^t_i \rangle$.

**2) User update:** After recommendation, each user $j \in [m]$ updates their feature vector $u^t_j$, based on which creator, say $i^t_j$, was recommended to them:

$$u^{t+1}_j = \mathcal{P}\big(u^t_j + \eta_u f(v^t_{i^t_j}, u^t_j) v^t_{i^t_j}\big). \tag{3}$$

Here, $\eta_u \in [0, 1]$ is a parameter controlling the rate of update, $f(v_i, u_j)$ is a function that quantifies the impact of creator $i$'s content on user $j$ (discussed in detail later), and $\mathcal{P}(x) = \frac{x}{\|x\|_2}$ is the projection back onto the unit sphere. Our user update model generalizes [14], which considers $u^{t+1}_j = \mathcal{P}(u^t_j + \eta_u \langle v^t_{i^t_j}, u^t_j \rangle v^t_{i^t_j})$, by replacing the inner product with a general function $f$.

**3) Creator update:** Creators also update their feature vectors based on which users are recommended their content. For each creator $i \in [n]$, let $J^t_i = \{j : i^t_j = i\}$ be the set of users being recommended creator $i$, then $v^t_i$ is updated by:

$$v^{t+1}_i = \mathcal{P}\Big(v^t_i + \frac{\eta_c}{|J^t_i|} \sum_{j \in J^t_i} g(u^t_j, v^t_i) u^t_j\Big), \tag{4}$$

where $\eta_c \in [0, 1]$ is a parameter controlling the rate of update, and $g(u_j, v_i)$ is a function that quantifies the impact of user $j$ on creator $i$.

**Impact functions $f$ and $g$**   Our results apply to any impact functions $f$ and $g$ that satisfy the following natural assumptions. First, $f(v_i, u_j)$ and the inner product $\langle v_i, u_j \rangle$ have the same sign: $f(v_i, u_j)$ is $\begin{cases} > 0 & \text{if } \langle v_i, u_j \rangle > 0 \\ < 0 & \text{if } \langle v_i, u_j \rangle < 0 \\ = 0 & \text{if } \langle v_i, u_j \rangle = 0. \end{cases}$ This means that if a user *likes* the content ($\langle v^t_i, u^t_j \rangle > 0$), then the

user vector $\boldsymbol{u}_j^t$ will be updated *towards* the direction of the creator vector $\boldsymbol{v}_j^t$. If the user *dislikes* the content ($\langle \boldsymbol{v}_i^t, \boldsymbol{u}_j^t \rangle < 0$), then the user vector $\boldsymbol{u}_j^t$ will move *away from* $\boldsymbol{v}_j^t$. Such "biased assimilation" user behavior is well documented in the literature [14]. Further, we assume upper and lower bounds on $|f|$:

$$|f(\boldsymbol{v}_i, \boldsymbol{u}_j)| \leq 1, \qquad |f(\boldsymbol{v}_i, \boldsymbol{u}_j)| \geq L_f > 0 \text{ whenever } \langle \boldsymbol{v}_i, \boldsymbol{u}_j \rangle \neq 0.$$

The lower bound $|f(\boldsymbol{v}_i, \boldsymbol{u}_j)| \geq L_f$ means that the exposure to an item that a user likes or dislikes always has some non-negligible impact on the user's preference. For example, $f(\boldsymbol{v}_i, \boldsymbol{u}_j) = \text{sign}(\langle \boldsymbol{v}_i, \boldsymbol{u}_j \rangle)a + b\langle \boldsymbol{v}_i, \boldsymbol{u}_j \rangle$ satisfies both assumptions when $L_f = a > 0$ and $b \geq 0$.

For $g$, likewise assume that its sign is the same as $\langle \boldsymbol{u}_j, \boldsymbol{v}_i \rangle$: $g(\boldsymbol{u}_j, \boldsymbol{v}_i)$ is $\begin{cases} > 0 & \text{if } \langle \boldsymbol{u}_j, \boldsymbol{v}_i \rangle > 0 \\ < 0 & \text{if } \langle \boldsymbol{u}_j, \boldsymbol{v}_i \rangle < 0 \\ = 0 & \text{if } \langle \boldsymbol{u}_j, \boldsymbol{v}_i \rangle = 0. \end{cases}$ Intuitively, this captures the incentive of a creator who aims to maximize the average ratings from users who are recommended their items. On video platforms for example, if the creators are rewarded based on the average rating of their videos, they will try to reinforce their creation styles based on the users who give positive feedback ($\langle \boldsymbol{u}_j, \boldsymbol{v}_i \rangle > 0$) so that their creations are more likely to be recommended to those users. Meanwhile, the creators will also change their creation styles based on negative feedback ($\langle \boldsymbol{u}_j, \boldsymbol{v}_i \rangle < 0$), but in the opposite direction of the negative-feedback users' interests, so that their creations are less likely to be recommended to those users. Taking both scenarios into account, the creator moves towards the weighted average of user preferences $\sum_{j \in J_i^t} g(\boldsymbol{u}_j^t, \boldsymbol{v}_i^t)\boldsymbol{u}_j^t$, which is captured by our update rule (4). A particular example of $g$ is the sign function $g(\boldsymbol{u}_j, \boldsymbol{v}_i) = \text{sign}(\langle \boldsymbol{u}_j, \boldsymbol{v}_i \rangle) \in \{-1, 0, 1\}$. We will only consider the sign function $g$ in order to simplify the theoretical presentation. We believe that all our results can be generalized to other $g$ functions satisfying similar conditions as $f$; the details are left as future work.

## 3 Unavoidable Polarization

Having defined the user-creator feature dynamics in a recommender system with dual influence, we now theoretically study how such dynamics evolve. Our main result is: if every creator can be recommended to every user with some non-zero probability, then the dynamics must eventually *polarize*.

**Definition 3.1** (consensus and bi-polarization). *Let $R > 0$. The dynamics $(\boldsymbol{U}^t, \boldsymbol{V}^t)$ is said to reach:*

- *$R$-consensus if there exists a vector $\boldsymbol{c} \in \mathbb{R}^d$ such that every feature vector is $R$-close to $\boldsymbol{c}$: $\forall \boldsymbol{u}_j^t, \|\boldsymbol{u}_j^t - \boldsymbol{c}\|_2 \leq R$ and $\forall \boldsymbol{v}_i^t, \|\boldsymbol{v}_i^t - \boldsymbol{c}\|_2 \leq R$.*

- *$R$-bi-polarization if there exists a vector $\boldsymbol{c} \in \mathbb{R}^d$ such that every feature vector is $R$-close to $+\boldsymbol{c}$ or $-\boldsymbol{c}$: $\forall \boldsymbol{u}_j^t, \|\boldsymbol{u}_j^t - \boldsymbol{c}\|_2 \leq R$ or $\|\boldsymbol{u}_j^t + \boldsymbol{c}\|_2 \leq R$, and $\forall \boldsymbol{v}_i^t, \|\boldsymbol{v}_i^t - \boldsymbol{c}\|_2 \leq R$ or $\|\boldsymbol{v}_i^t + \boldsymbol{c}\|_2 \leq R$.*

*The dynamics is said to reach $(R, \boldsymbol{c})$-consensus (or $(R, \boldsymbol{c})$-bi-polarization) if the dynamics reaches $R$-consensus (or $R$-bi-polarization) with the vector $\boldsymbol{c}$.*

Consensus is any state where all users and creators have similar feature vectors (with maximum difference $R$), implying that they have similar interests or preferences. Bi-polarization is any state where all users and creators are clustered into two groups with exactly opposite features (e.g., Republicans vs Democrats). Mathematically, consensus is a special case of bi-polarization.

**Proposition 3.2.** *Bi-polarization states are absorbing: once the dynamics reaches $(R, \boldsymbol{c})$-bi-polarization with some $R \in [0, 1]$ and $\boldsymbol{c} \in \mathbb{S}^{d-1}$, it will satisfy $(R, \boldsymbol{c})$-bi-polarization forever. The same holds for consensus.*

A natural property of a recommender system is that every creator can be recommended to every user with some non-zero probability: $p_{ij}^t \geq p_0 > 0$ with some constant $p_0$. This is satisfied by the softmax function, which is a rough model of real-world recommendation algorithms [13, 26]: $p_{ij}^t = \frac{\exp(\beta \langle \boldsymbol{u}_j^t, \boldsymbol{v}_i^t \rangle)}{\sum_{i=1}^n \exp(\beta \langle \boldsymbol{u}_j^t, \boldsymbol{v}_i^t \rangle)} \geq \frac{\exp(-\beta)}{n \exp(\beta)} = p_0 > 0$. Moreover, many large-scale real-world recomendation systems (e.g., Yahoo! [28] and Kuaishou [16]) intentionally insert small random traffic attempting to improve recommendation diversity or explore users' interests [25, 40], which will cause all recommendation probabilities to be non-zero. We show in Theorem 3.3 that, however, a recommender system satisfying $p_{ij}^t \geq p_0 > 0$ must converge to polarization, under some additional conditions on the users' and creators' update rates:

**Theorem 3.3.** *Suppose $g(\boldsymbol{u}_j, \boldsymbol{v}_i) = \text{sign}(\langle \boldsymbol{u}_j, \boldsymbol{v}_i \rangle)$, the update rates $\eta_c \leq \frac{\eta_u L_f}{2}$ and $\eta_u < \frac{1}{2}$, and the recommendation probability $p_{ij}^t \geq p_0 > 0, \forall i, j, t$. Then, from almost all initial states, the dynamics $(\boldsymbol{U}^t, \boldsymbol{V}^t)$ will eventually reach R-consensus or R-bi-polarization for any $R > 0$.*

In other words, if the users' and creators' updates are not too fast and all recommendation probabilities are non-zero, then all users and creators will eventually converge to at most two clusters (regardless of the feature dimension $d$). Since creators in one cluster produce similar contents, users in such a polarized system can never receive diverse recommendations. This means that the naïve attempt of imposing $p_{ij}^t \geq p_0 > 0$ cannot improve the diversity of a recommender system with dual influence. The conditions on the update rates $\eta_u, \eta_c$ are only assumed to simplify the proof of Theorem 3.3. Our experiments (in Section 5) will show that polarization still occurs even without those conditions.

Theorem 3.3 does not characterize the rate of convergence of the user-creator feature dynamics to polarization, which we leave as an open question.

The proof of Theorem 3.3 is an absorbing Markov chain argument. It uses the following lemma:

**Lemma 3.4.** *Suppose $\eta_c \leq \frac{\eta_u L_f}{2}$ and $\eta_u < \frac{1}{2}$. For any $R > 0$, for almost every state $(\boldsymbol{U}^t, \boldsymbol{V}^t)$ in the state space, there exists a path $(\boldsymbol{U}^t, \boldsymbol{V}^t) \to (\boldsymbol{U}^{t+1}, \boldsymbol{V}^{t+1}) \to \cdots \to (\boldsymbol{U}^{t+T}, \boldsymbol{V}^{t+T})$ of finite length that leads to an R-bi-polarization state $(\boldsymbol{U}^{t+T}, \boldsymbol{V}^{t+T})$.*

The proof of this lemma (in Appendix F) is involved. It uses induction on the number of creators $n$. The base case of $n = 1$ is proved by a potential function argument. For $n \geq 2$, we first construct a path that leads the *subsystem* of $n-1$ creators and all users to $R$-bi-polarization. Then, depending on where the remaining creator is, we construct a sequence of recommendations that leads the remaining creator to one of the two clusters formed by the $n-1$ creators and all users. Such recommendations will move some users out of the formed clusters, which requires extra care in the proof.

*Proof of Theorem 3.3.* For any state $(\boldsymbol{U}^t, \boldsymbol{V}^t)$ in the state space, by Lemma 3.4 there exists a path $(\boldsymbol{U}^t, \boldsymbol{V}^t) \to \cdots \to (\boldsymbol{U}^{t+T}, \boldsymbol{V}^{t+T})$ of length $T$ that leads to $R$-bi-polarization. Because every creator can be recommended to a user with probability at least $p_0$, each transition $(\boldsymbol{U}^{t'}, \boldsymbol{V}^{t'}) \to (\boldsymbol{U}^{t'+1}, \boldsymbol{V}^{t'+1})$ happens with probability at least $p_0^m$. So, the path of length $T$ has probability at least $p_0^{mT} > 0$, and the probability that the dynamics *does not* reach $R$-bi-polarization after $KT$ steps is at most $(1 - p_0^{mT})^K$, which $\to 0$ as $K \to \infty$. Therefore, with probability 1 the dynamics will reach $R$-bi-polarization eventually. $\square$

## 4 Discussions on Real-World Designs

Next, we discuss how 4 types of real-world recommender system designs affect the user-creator feature dynamics: top-$k$ truncation, threshold truncation, diversity-boosting, and uniform traffic.

**(1) Top-$k$ Truncation** A prevalent practice in modern two-stage recommendation algorithms on large-scale platforms, such as YouTube [13], is to first filter out items that are unlikely to be relevant to a user, then make recommendations from the remaining items. In particular, we consider the top-$k$ truncation policy: for every user $j$, find the $k$ most relevant creators, namely, the $k$ creators whose inner products with the user $\langle \boldsymbol{v}_i^t, \boldsymbol{u}_j^t \rangle$ are largest (equivalently, the $k$ creators whose probabilities $p_{ij}^t$ of being recommended to user $j$ are highest), then recommend one of those $k$ creators to user $j$ with probability proportional to $p_{ij}^t$. The other creators will not be recommended. This practice significantly reduces the computation cost and improves the relevancy of recommendations. Interestingly, we show that such a practice also has the potential to improve the long-term diversity of a recommender system with dual influence.

**Definition 4.1** (clusters). *We say a state $(\boldsymbol{U}^t, \boldsymbol{V}^t)$ forms $q$ clusters if there exist $\boldsymbol{c}_1, \ldots, \boldsymbol{c}_q \in \mathbb{R}^d$ and a small number $R > 0$ such that every feature vector is in the $\ell_2$ ball of some $\boldsymbol{c}_i$ with radius $R$ (denoted by $B(\boldsymbol{c}_\ell, R) = \{\boldsymbol{x} : \|\boldsymbol{x} - \boldsymbol{c}_\ell\|_2 \leq R\}$), and $B(\boldsymbol{c}_\ell, 2R) \cap B(\boldsymbol{c}_{\ell'}, 2R) = \emptyset$ for $\ell \neq \ell'$.*

It is clear that consensus has a single cluster, and bi-polarization has two.

**Proposition 4.2.** *With top-$k$ truncation, there exist states $(\boldsymbol{U}^t, \boldsymbol{V}^t)$ that form $\lfloor n/k \rfloor$ clusters and are absorbing (i.e., once the system forms $\lfloor n/k \rfloor$ clusters, it forms $\lfloor n/k \rfloor$ clusters forever).*

This result is in contrast with Theorem 3.3 which shows that a recommender system where every creator can be recommended to every user ($p_{ij}^t > 0$) is doomed to polarize. With top-$k$ truncation where some $p_{ij}^t = 0$, polarization can be avoided. Experiments in Section 5.3 support our prediction that top-$k$ truncation can reduce polarization and improve diversity.

**(2) Threshold Truncation**   Besides top-$k$ truncation, threshold truncation is another way to filter out irrelevant creators: set a threshold $\tau \in [-1, 1]$ such that any user-creator pair with inner product $\langle \boldsymbol{u}_i, \boldsymbol{v}_j \rangle < \tau$ is not recommended. A natural choice is $\tau = 0$, meaning that users will not receive recommendations predicted to be "disliked" by them. Increasing $\tau$ is similar to increasing the $\beta$ in the softmax function, which improves recommendation relevance.

**Proposition 4.3.** *In $d$-dimensional feature space, if user-creator pairs with $\langle \boldsymbol{u}_i, \boldsymbol{v}_j \rangle < 0$ are not recommended, then there exist stable states with $d + 1$ clusters.*

Although truncation at $\tau = 0$ allows stable states with $d + 1$ clusters to exist, the dynamics does not necessarily converge to such states; it can still end up with stable states with fewer clusters. In fact, experiments (in Appendix B.2) show that truncation at $\tau = 0$ is *not good* for diversity and causes severe polarization, while truncation at a large threshold like $\tau = 0.707$ is better at reducing polarization.

**(3) Diversity Boosting**   Diversity boosting aims to explore users' interests and improve users' experience by diversifying recommendation. For example, when making recommendations, the model optimizes the objective:

$$h_{rel}(\langle \boldsymbol{u}_i, \boldsymbol{v}_j \rangle) + \rho h_{div}(list_i, \boldsymbol{v}_j), \tag{5}$$

where $h_{rel}, h_{div}$ rewards the recommendation relevance and diversity respectively and $list_i$ records the recent list of recommended items to user $i$. $h_{div}$ can take a simple form of $\sum_{j' \in list_i} 1 - \langle \boldsymbol{v}_{j'}, \boldsymbol{v}_j \rangle$, and $\rho > 0$ controls the strength of diversity-boosting. Despite being successful when users' preferences and items are fixed, this design alone cannot prevent bi-polarization in our dual-influence dynamics, since the conditions in Theorem 3.3 are still satisfied and the users' and creators' update rules remain the same. Experiments in Section 5.2 support our claim.

**(4) Uniform Traffic**   Adding a small fraction of uniform traffic to the personalized recommendations is another method proposed in previous works to improve recommendation diversity or to explore user preferences [25, 16, 9, 8, 30]. This method gives a non-zero lower bound on the probability of every creator being recommended to every user. So, as a corollary of our Theorem 3.3, it causes a recommender system with dual influence to polarize. Such an observation is striking as it demonstrates that optimizing for recommendation diversity in a static setting can ultimately lead to a huge loss of the system diversity in the long run.

## 5   Experiments

We present experimental results on the behavior of user-creator feature dynamics on synthetic data and real-world (MovieLens 20M) data and the effect of top-$k$ truncation and threshold truncation on the dynamics.

### 5.1   Synthetic Data Experiments

**Setup**   The dynamics is initialized by randomly generating user and creator features on the unit sphere in $\mathbb{R}^d$. We pick $d = 10$, number of creators $n = 50$, number of users $m = 100$. We use the softmax recommendation probability function (2). We simulate the dynamics for $T = 1000$ steps, repeated 100 times each with a new initialization. We choose the sign impact function $g(\boldsymbol{u}_j, \boldsymbol{v}_i) = \text{sign}(\langle \boldsymbol{u}_j, \boldsymbol{v}_i \rangle)$ for creator updates. For user updates, we choose inner product $f(\boldsymbol{v}_i, \boldsymbol{u}_j) = \langle \boldsymbol{v}_i, \boldsymbol{u}_j \rangle$. The inner product function is studied in previous works on users' preference dynamics (but not creators') [14]. Note that the inner product does not satisfy the condition $|f(\boldsymbol{v}_i, \boldsymbol{u}_j)| \geq L_f$ needed in Theorem 3.3. However, we still observe convergence to polarization in nearly all experiments. Thus, even when this condition does not hold, users and creators still tend towards polarization in practice.

Three key parameters in our model are $\beta$ (sensitivity of the softmax function), $\eta_c$ (creator update rate), and $\eta_u$ (user update rate). We set them to $\beta = 1, \eta_c = \eta_u = 0.1$, and change one parameter at

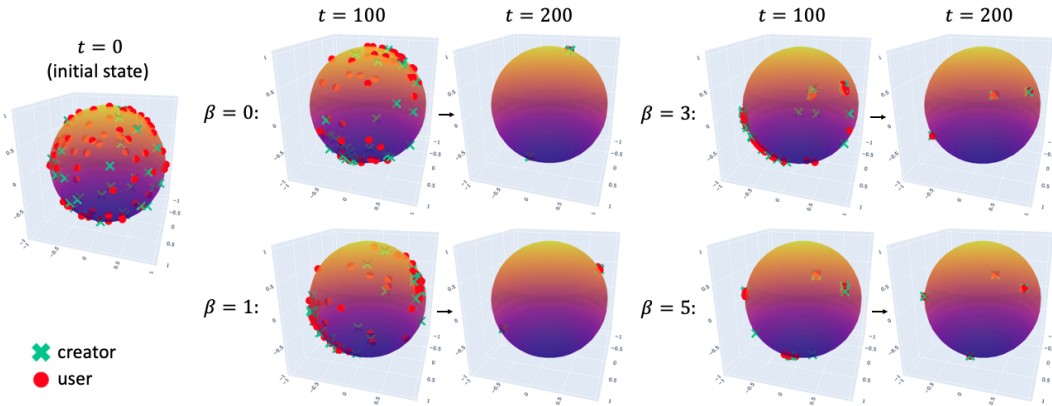

Figure 1: Snapshots of the dynamics simulated with the same initialization but different recommendation sensitivity $\beta$. A larger $\beta$ resulted in more clusters at time step $t = 200$.

a time to see its effect on the dynamics. We also test what happens when some dimensions of the user features are *fixed* features that are not updated.

**Measures**    To quantify the behavior of the dynamics, given user and creator feature vectors $(U, V)$ we compute the following measures, which cover diversity, relevancy, and polarization of the system:

- *Creator Diversity* (CD): diversity of the creator features, measured by their average pairwise distance [47, 33]: $\mathrm{CD}(V) = \frac{1}{n(n-1)} \sum_{i=1}^{n} \sum_{j \neq i} \|v_i - v_j\|$.

- *Recommendation Diversity* (RD): diversity of the contents recommended to a user, measured by the weighted variance of the contents: $\mathrm{RD}(U, V; \beta) = \frac{1}{m} \sum_{j=1}^{m} \sum_{i=1}^{n} p_{ij} \|v_i - \overline{v}_j\|^2$, where $\overline{v}_j = \sum_{i=1}^{n} p_{ij} v_i$ and $p_{ij} = \frac{\exp(\beta \langle u_j, v_i \rangle)}{\sum_{i=1}^{n} \exp(\beta \langle u_j, v_i \rangle)}$.

- *Recommendation Relevance* (RR): relevance of the contents recommended to a user, measured by the weighted average of inner products: $\mathrm{RR}(U, V; \beta) = \frac{1}{m} \sum_{j=1}^{m} \sum_{i=1}^{n} p_{ij} \langle u_j, v_i \rangle$.

- *Tendency to Polarization* (TP): This is a novel measure we propose to quantify how close the system is to consensus or bi-polarization, measured by the average absolute inner products between the creators: $\mathrm{TP}(V) = \frac{1}{n^2} \sum_{i=1}^{n} \sum_{k=1}^{n} |\langle v_i, v_k \rangle|$. $\mathrm{TP}(V)$ being closer to 1 means that the system is more polarized, because the term $|\langle v_i, v_k \rangle|$ is 1 iff the two vectors $v_i, v_k$ are equal or opposite to each other.

It is worth noting that a high creator diversity is necessary for simultaneously achieving high recommendation relevance and high recommendation diversity. For example, they cannot be simultaneously achieved in a polarized state.

**Sensitivity Parameter $\beta$**    A larger $\beta$ means that a user will be recommended more relevant content/creator with a higher probability. $\beta = 0$, on the other hand, means that the user receives uniform recommendations across all creators. **Our main observation** from the experiments is: *a larger $\beta$ leads to higher creator diversity and alleviated polarization in the long run.*

Figure 1 shows snapshots of the dynamics at different time steps under different $\beta$ values. Here, we choose dimension $d = 3$ instead of 10 so the feature vectors can be visualized on a 3d sphere. We see that the system tends to form more clusters at time $t = 200$ as $\beta$ increases.

Figure 2 shows the changes of the 4 measures CD, RD, RR, TP over time under different $\beta$ values. We see that a more diverse recommendation policy (a smaller $\beta$) leads to lower creator diversity and a higher level of polarization in the long run. In particular, while Creator Diversity reaches a similar level under different $\beta$ in the end, it *drops at a slower rate* with a *larger $\beta$* (see $\beta = 5, 6$). Moreover, from the plot of Tendency to Polarization, we see that a larger $\beta$ *alleviates* polarization, which means improvement in the diversity of the whole system.

An explanation for our observation is the following: When $\beta$ is smaller, each user receives more uniform recommendations across all creators. So, for different creators, the sets of users recommended

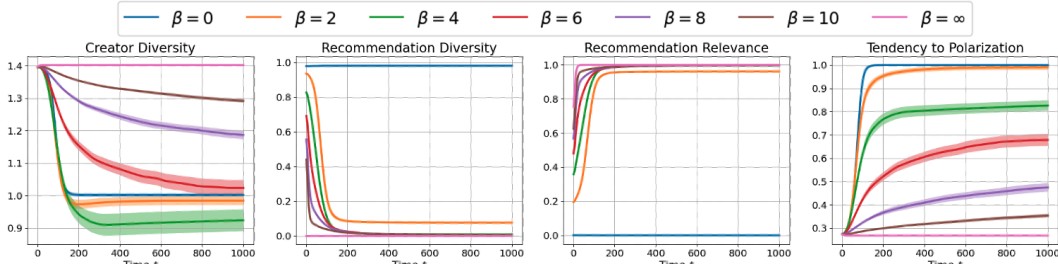

Figure 2: Changes of measures over time under different sensitivity parameter $\beta$, on synthetic data. $\beta = 0$ means uniform (non-personalized) recommendation. $\beta = \infty$ means hard-max recommendation: only recommend the single most relevant creator to a user. Larger $\beta$ reduces the tendency to polarization.

to those creators have larger intersections. Since the creator updates are based on the sets of recommended users, different creators will be moving towards more similar directions. This leads to faster polarization. One can also predict this observation from Theorem 3.3: when $\beta$ is large, the minimum recommendation probability $p_0$ of the softmax function tends to 0, so it might take a long time for the system to converge to polarization, while with a small $\beta$ the system polarizes quickly.

**Update Rates** $\eta_c$ **and** $\eta_u$ A larger $\eta_c$ means that creator features are updated faster, and intuitively should lead to faster polarization. This is validated in experiments: Figure 3 shows that a larger $\eta_c$ indeed causes more extreme polarization and lower diversity (both CD and RD). A larger $\eta_u$ means that user features are updated faster. It has a similar effect of exacerbating polarization as $\eta_c$ does, as shown in Figure 4.

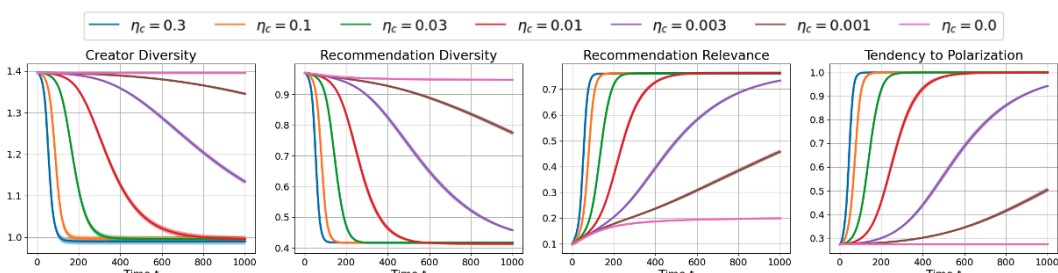

Figure 3: Changes of measures over time under different creator update rate $\eta_c$, on synthetic data

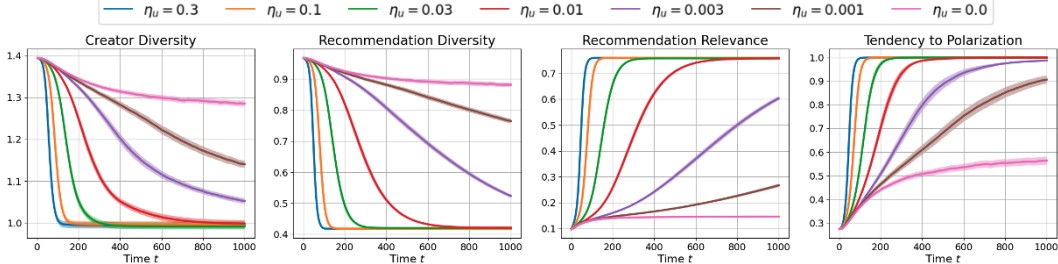

Figure 4: Changes of measures over time under different user update rate $\eta_u$, on synthetic data

**Number of Fixed Dimensions** We also consider the scenario where some dimensions of the user feature vectors are fixed features and thus not updated from round to round (e.g., age, gender), which is a realistic scenario. Detailed results are in Appendix B.1. The **main observation** is: *as the number of fixed dimensions increases, the diversity of the system improves and the degree of polarization is reduced.* This is similar to the effect of decreasing user update rate $\eta_u$. The observation that

fixed dimensions of user features help to improve diversity might be a reason why the recommender systems in practice are not as polarized as our theoretical prediction.

## 5.2 Real-World Data Experiments

In this part, we conduct experiments on the MovieLens 20M dataset [19]. We use a real-world two-tower recommendation model with 16-dimensional tower tops as the user and creator embeddings (Figure 9). The model is initialized by fitting a two-tower model [21] on the existing MovieLens rating data and using the tower tops as the initial user and creator embeddings. Then we follow Algorithm 1 to simulate the dynamics.

Figure 5 shows the effect of the recommendation sensitivity parameter $\beta$ on the system. Similar to the synthetic data experiments, a smaller $\beta$ (more diverse recommendation for the users in the short term) results in faster polarization. We note that the joint results on CD and TP are more informative than each one alone: despite $\beta = 0$ has a higher creator diversity than $\beta = 2$ at $T = 500$, the system reaches polarization more quickly under $\beta = 0$. The higher creator diversity under $\beta = 0$ is because the two clusters in the bi-polarized state are more balanced so the average pairwise distance between the creators is higher under $\beta = 0$ than under $\beta = 2$.

Figure 6 shows the effect of using diversity-aware objective (Eq. 5) for diversity boosting. We see that myopically promoting the short-term recommendation diversity (using a larger $\rho$) results a higher creation diversity but also a higher tendency to polarization. A possible explanation for this phenomenon is, similar to the case with $\beta$, the system polarizes into two balanced clusters which actually have a large average pairwise distance. In this case, Tendency to Polarization is a better measure for diversity loss than Creator Diversity (average pairwise distance).

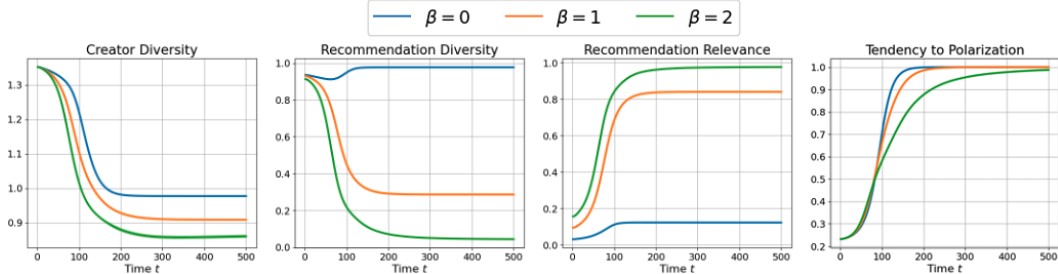

Figure 5: Experiment on MovieLens 20M dataset under different recommendation sensitivity $\beta$

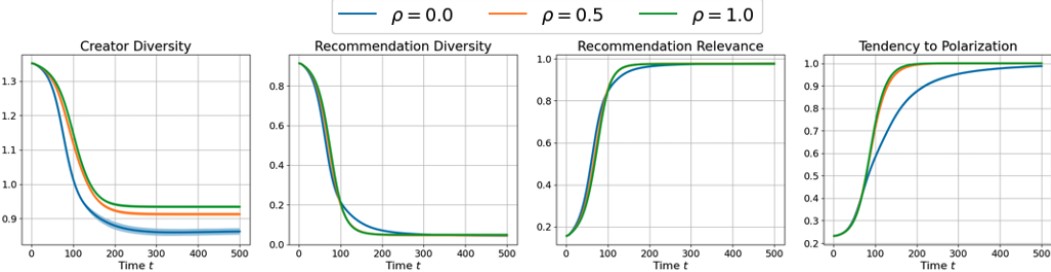

Figure 6: Experiment on MovieLens 20M dataset with diversity-aware objective under different $\rho$

## 5.3 Top-$k$ Truncation and Threshold Truncation

We experimented with top-$k$ truncation on the synthetic data (Table 1) and the MovieLens dataset (Appendix C). Our **main observation** is: *a small $k$ improves the diversity of the recommender system and reduces polarization*. This is consistent with our theoretical prediction (Proposition 4.2). However, there is a tradeoff between the diversity of recommendations to users (RD) and the diversity of creations in the system (CD and TP). A top-$k$ truncation policy with small $k$ is "not diverse" for users because it exposes a user only to a small set of contents. However, such a policy can lead to a more diverse outcome in the whole system. This tradeoff is worth further studying.

Table 1: Diversity improvement by top-$k$ truncation on synthetic data

| $\beta$ | $k$ | Creator Diversity | Recommendation Diversity | Recommendation Relevance | Tendency to Polarization |
|---|---|---|---|---|---|
| 1 | 50 | $1.00_{\pm.03}$ | $\mathbf{0.42_{\pm 0.01}}$ | $0.76_{\pm 0.01}$ | $1.00_{\pm 10^{-3}}$ |
| | 25 | $0.52_{\pm.32}$ | $0.03_{\pm 0.03}$ | $0.97_{\pm 0.02}$ | $0.91_{\pm 0.13}$ |
| | 20 | $0.91_{\pm.15}$ | $0.00_{\pm 0.01}$ | $1.00_{\pm 0.01}$ | $0.68_{\pm 0.12}$ |
| | 10 | $1.17_{\pm.06}$ | $0.00_{\pm 10^{-3}}$ | $1.00_{\pm 10^{-3}}$ | $0.50_{\pm 0.07}$ |
| | 5 | $1.31_{\pm.02}$ | $0.00_{\pm 10^{-3}}$ | $1.00_{\pm 10^{-3}}$ | $0.35_{\pm 0.03}$ |
| | 1 | $\mathbf{1.40_{\pm 10^{-3}}}$ | $0.00_{\pm 10^{-3}}$ | $\mathbf{1.00_{\pm 10^{-3}}}$ | $\mathbf{0.27_{\pm 10^{-3}}}$ |
| 3 | 50 | $0.95_{\pm.14}$ | $\mathbf{0.02_{\pm 0.02}}$ | $0.99_{\pm 0.01}$ | $0.91_{\pm 0.10}$ |
| | 25 | $0.80_{\pm.24}$ | $0.00_{\pm 0.01}$ | $1.00_{\pm 10^{-3}}$ | $0.77_{\pm 0.13}$ |
| | 20 | $0.89_{\pm.13}$ | $0.00_{\pm 10^{-3}}$ | $1.00_{\pm 10^{-3}}$ | $0.74_{\pm 0.11}$ |
| | 10 | $1.18_{\pm.05}$ | $0.00_{\pm 10^{-3}}$ | $1.00_{\pm 10^{-3}}$ | $0.49_{\pm 0.07}$ |
| | 5 | $1.31_{\pm.02}$ | $0.00_{\pm 10^{-3}}$ | $1.00_{\pm 10^{-3}}$ | $0.34_{\pm 0.03}$ |
| | 1 | $\mathbf{1.40_{\pm 10^{-3}}}$ | $0.00_{\pm 10^{-3}}$ | $\mathbf{1.00_{\pm 10^{-3}}}$ | $\mathbf{0.27_{\pm 10^{-3}}}$ |

We also experimented with threshold truncation on synthetic data (Appendix B.2) and MovieLens data (Appendix C). The effect of a large truncation threshold $\tau$ is similar to the effect of a small $k$ in top-$k$ truncation.

## 6  Conclusion

Our work defines a dynamics model to capture the dual influence of recommender systems on user preferences and content creation. Although our model is a theoretical abstraction, we believe that it captures the essence of a real-world recommender system, and our effort is an important initial endeavor to study diversity in recommender systems with dual influence. (See Appendix H for some additional discussions on real-world recommender systems.) We specifically point out different concepts of diversity in recommender systems (creation diversity, recommendation diversity, and tendency to polarization) and provide theoretical and empirical evidences to show that, due to dual influence, myopically optimizing recommendation diversity might hurt the long-term creation diversity and result in polarization of the system. We also explore popular design choices in recommender systems and show an interesting and somewhat counter-intuitive result that designs purely targeting efficiency improvement (e.g., top-$k$ truncation) can alleviate polarization. We believe that the insights from our work are valuable to building healthy and sustainable recommender systems, and our results can inspire more sophisticated solutions for improving the long-term diversity of recommender systems to be developed.

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

# A  Additional Discussions on Related Works

Table 2: Comparison between our work and some previous works on performative effects of recommender systems

| Works | Adaptive Users? | Adaptive Creators? | Creator Reward | Dynamics or Equilibrium? | Content Adjustment Model |
|---|---|---|---|---|---|
| Ours | Yes | Yes | User engagement | Dynamics | Conditioned on previous time step; implicit cost of content adjustment |
| [15] | No | Yes | Exposure | Dynamics | Conditioned on previous time step; explicit cost of content adjustment |
| [42] | No | Yes | User engagement | Dynamics | Freely choose without cost |
| [35] | No | Yes | User engagement | Dynamics | Freely choose without cost |
| [23] | No | Yes | Exposure | Equilibrium | Freely choose with cost |
| [20] | No | Yes | Exposure | Equilibrium | Freely choose without cost |
| [7] | No | Yes | Exposure | Equilibrium | Freely choose without cost |
| [2] | No | Yes | User engagement | Equilibrium | Freely choose without cost |
| [43] | No | Yes | Designed by a welfare-maximizing platform | Dynamics | Freely choose without cost |
| [14] | Yes | No[1] | N/A | Dynamics | N/A |
| [41] | Yes | No[1] | N/A | Dynamics | N/A |
| [3] | Adaptive and adversarial | No[1] | N/A | Dynamics | N/A |

[1]: These works study the design of recommendation algorithms for the platform with a fixed set of content, without explicitly modeling the content creators.

# B  Additional Experiments on Synthetic Data

## B.1  Number of Fixed Dimensions

In this part, we consider the case where some dimensions of the user feature vectors are fixed features and thus not updated from round to round (e.g., ages, genders). Formally, we fix the first $k \leq d$ dimensions. The remaining $d - k$ dimensions $\boldsymbol{u}_j^t[k + 1 : d] = (u_j^t[k + 1], \ldots, u_j^t[d])$ are updated according to the following rule: $\boldsymbol{u}_j^{t+1}[k + 1 : d] = \|\boldsymbol{u}_j^t[k + 1 : d]\| \cdot \mathcal{P}\big(\boldsymbol{u}_j^t[k + 1 : d] + \eta_u f(\boldsymbol{v}_i^t, \boldsymbol{u}_j^t)\boldsymbol{v}_i^t[k+1 : d]\big)$. The multiplication by $\|\boldsymbol{u}_j^t[k + 1 : d]\|$ ensures unit norm $\|\boldsymbol{u}_j^{t+1}\| = 1$. The effect of the number of fixed dimensions on the dynamics is shown in Figure 7. We see that the diversity of the system *improves* as the number of fixed dimensions increases, and the degree of polarization is reduced. This is similar to the effect of decreasing user update rate $\eta_u$ in Figure 4. The observation that fixed user features encourage diversity might be a reason why the recommender systems in practice are not as polarized as our theoretical prediction.

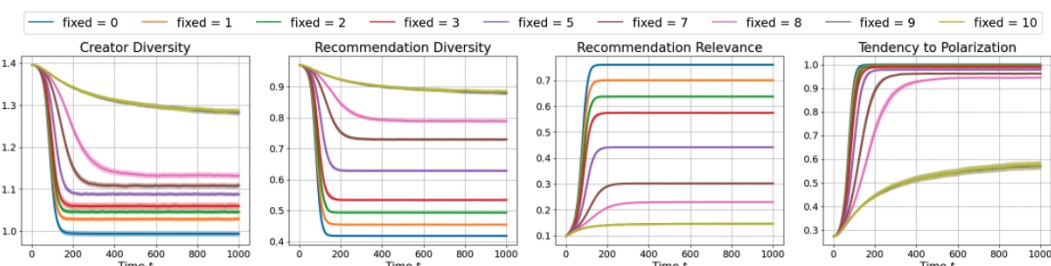

Figure 7: Changes of measures over time under different numbers of fixed dimensions, on synthetic data

## B.2 Threshold Truncation

Table 3 shows that the effect of different thresholds in threshold truncation on the long-term diversity of the system. We see that truncating at $\tau = 0$, which corresponds to $90°$ angle between $\boldsymbol{u}_j$ and $\boldsymbol{v}_i$, is *not good* for diversity, resulting in the lowest creator diversity measure (CD) and highest tendency to polarization (TP). Truncating at a large threshold like $0.707$ is good for diversity, instead.

Figure 8 shows how the diversity measures change over time, under different truncation thresholds.

Table 3: Diversity improvement by threshold truncation on synthetic data

| $\beta$ | threshold $\tau$ | CD | RD | RR | TP |
|---|---|---|---|---|---|
| | $-\cos(60°) = -0.5$ | $1.00 \pm 0.03$ | $0.00 \pm 10^{-3}$ | $1.00 \pm 10^{-3}$ | $0.99 \pm 10^{-3}$ |
| | $-\cos(72°) = -0.309$ | $0.96 \pm 0.06$ | $\mathbf{0.01 \pm 0.02}$ | $1.00 \pm 0.02$ | $0.92 \pm 0.10$ |
| | $\cos(90°) = 0$ | $0.03 \pm 0.16$ | $0.00 \pm 10^{-3}$ | $1.00 \pm 10^{-3}$ | $0.99 \pm 0.04$ |
| 0 | $\cos(72°) = 0.309$ | $0.72 \pm 0.30$ | $0.00 \pm 10^{-3}$ | $1.00 \pm 10^{-3}$ | $0.81 \pm 0.12$ |
| | $\cos(60°) = 0.5$ | $1.16 \pm 0.11$ | $0.00 \pm 10^{-3}$ | $1.00 \pm 10^{-3}$ | $0.47 \pm 0.10$ |
| | $\cos(45°) = 0.707$ | $\mathbf{1.37 \pm 0.02}$ | $0.00 \pm 10^{-3}$ | $\mathbf{1.00 \pm 10^{-3}}$ | $\mathbf{0.33 \pm 0.02}$ |
| | $\cos(30°) = 0.866$ | $1.30 \pm 0.03$ | $0.00 \pm 10^{-3}$ | $1.00 \pm 10^{-3}$ | $0.55 \pm 0.05$ |
| | $-\cos(60°) = -0.5$ | $0.98 \pm 0.04$ | $0.00 \pm 0.02$ | $1.00 \pm 0.01$ | $0.96 \pm 0.04$ |
| | $-\cos(72°) = -0.309$ | $0.92 \pm 0.08$ | $0.00 \pm 0.02$ | $0.99 \pm 0.02$ | $0.87 \pm 0.10$ |
| | $\cos(90°) = 0$ | $0.13 \pm 0.31$ | $0.00 \pm 10^{-3}$ | $1.00 \pm 10^{-3}$ | $0.97 \pm 0.08$ |
| 1 | $\cos(72°) = 0.309$ | $0.85 \pm 0.16$ | $0.00 \pm 10^{-3}$ | $1.00 \pm 10^{-3}$ | $0.76 \pm 0.11$ |
| | $\cos(60°) = 0.5$ | $1.21 \pm 0.07$ | $0.00 \pm 10^{-3}$ | $1.00 \pm 10^{-3}$ | $0.43 \pm 0.08$ |
| | $\cos(45°) = 0.707$ | $\mathbf{1.38 \pm 0.01}$ | $0.00 \pm 10^{-3}$ | $\mathbf{1.00 \pm 10^{-3}}$ | $\mathbf{0.30 \pm 0.01}$ |
| | $\cos(30°) = 0.866$ | $1.33 \pm 0.02$ | $0.00 \pm 10^{-3}$ | $1.00 \pm 10^{-3}$ | $0.47 \pm 0.04$ |
| | $-\cos(60°) = -0.5$ | $0.91 \pm 0.18$ | $\mathbf{0.01 \pm 0.02}$ | $1.00 \pm 0.01$ | $0.83 \pm 0.10$ |
| | $-\cos(72°) = -0.309$ | $0.85 \pm 0.23$ | $0.00 \pm 10^{-3}$ | $1.00 \pm 10^{-3}$ | $0.78 \pm 0.11$ |
| | $\cos(90°) = 0$ | $0.64 \pm 0.33$ | $0.00 \pm 10^{-3}$ | $1.00 \pm 10^{-3}$ | $0.81 \pm 0.12$ |
| 3 | $\cos(72°) = 0.309$ | $1.01 \pm 0.14$ | $0.00 \pm 10^{-3}$ | $1.00 \pm 10^{-3}$ | $0.64 \pm 0.14$ |
| | $\cos(60°) = 0.5$ | $1.26 \pm 0.05$ | $0.00 \pm 10^{-3}$ | $1.00 \pm 10^{-3}$ | $0.38 \pm 0.06$ |
| | $\cos(45°) = 0.707$ | $\mathbf{1.39 \pm 0.01}$ | $0.00 \pm 10^{-3}$ | $\mathbf{1.00 \pm 10^{-3}}$ | $\mathbf{0.28 \pm 0.01}$ |
| | $\cos(30°) = 0.866$ | $1.37 \pm 0.01$ | $0.00 \pm 10^{-3}$ | $1.00 \pm 10^{-3}$ | $0.34 \pm 0.01$ |

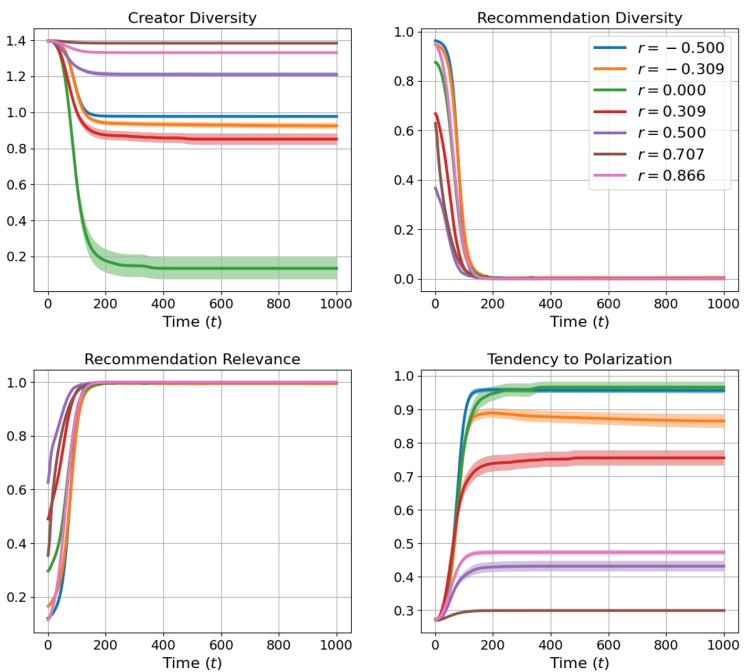

Figure 8: Changes of measures over time under different truncation threshold $\tau$, with $\beta = 1$, on synthetic data

# C  Additional Experiments on Real-World Dataset

## C.1  Details of the Recommendation Algorithm

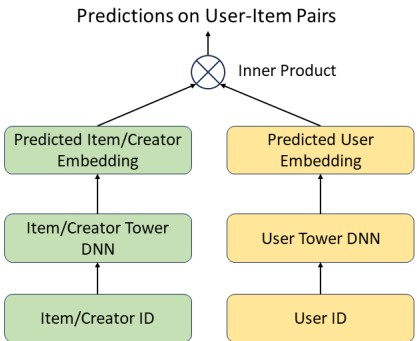

Figure 9: Two tower model for the MovieLens experiment, where the two towers both have size $16 \times 16$ with linear layers and ReLu activations.

---

**Algorithm 1** Real-world Recommendation with Dual Influence

---

**Input:** $t = 0$, actual embedding $U^{(0)}, V^{(0)}$, true labels $Y_{ij}^{(0)} := y(u_i^{(0)}, v_j^{(0)})$, initial parameter $\boldsymbol{\theta}^{(0)}$ (which includes the predicted embedding $\hat{U}^{(0)}, \hat{V}^{(0)}$)
**repeat**
    Let temporary parameter $\boldsymbol{w}^{(0)} \leftarrow \boldsymbol{\theta}^{(t)}$
    Compute loss $\mathcal{L}(\boldsymbol{\theta}^{(t)}, Y^{(t)})$
    **for** $s = 1$ **to** $m - 1$ **do**
        $\boldsymbol{w}^{(s+1)} \leftarrow \boldsymbol{\theta}^{(s)} - \eta \nabla_{\boldsymbol{w}} \mathcal{L}(\boldsymbol{w}^{(s)}, Y^{(t)})$
    **end for**
    $\boldsymbol{\theta}^{(t+1)} \leftarrow \boldsymbol{w}^{(m)}$
    Deliver recommendations based on $\hat{U}^{(t+1)}, \hat{V}^{(t+1)}$
    Update $U^{(t+1)}, V^{(t+1)}$, and $Y^{(t+1)}$
    $t \leftarrow t + 1$
**until** $\|\boldsymbol{\theta}^{(t)} - \boldsymbol{\theta}^{(t-1)}\|_2 \leq \delta$

---

## C.2  Top-$k$ and Threshold Truncations

We also try top-$k$ and threshold truncations (Section 4) on the MovieLens 20M dataset. Here, we have $n = 2000$ creators and $m = 2000$ users, with feature dimension $d = 16$. The results for top-$k$ truncation are in Table 4 and Figure 10. Similar to the experiments with synthetic data, we see that a smaller $k$ improves Creator Diversity (CD) and Recommendation Relevance (RR), reduces Tendency to Polarization (TP), yet worsens Recommendation Diversity (RD).

Table 4: Diversity improvement by top-$k$ truncation on MovieLens 20M dataset

| $\beta$ | $k$ | CD | RD | RR | TP |
|---|---|---|---|---|---|
| 0 | 2000 | $1.00 \pm 10^{-3}$ | $\mathbf{1.00 \pm 10^{-3}}$ | $0.00 \pm 10^{-3}$ | $1.00 \pm 10^{-3}$ |
| | 1000 | $0.30 \pm 0.04$ | $0.03 \pm 0.01$ | $0.88 \pm 0.01$ | $1.00 \pm 10^{-3}$ |
| | 500 | $1.10 \pm 0.06$ | $0.00 \pm 10^{-3}$ | $1.00 \pm 10^{-3}$ | $0.43 \pm 0.03$ |
| | 100 | $1.36 \pm 10^{-3}$ | $0.00 \pm 10^{-3}$ | $1.00 \pm 10^{-3}$ | $0.28 \pm 0.01$ |
| | 10 | $1.40 \pm 10^{-3}$ | $0.00 \pm 10^{-3}$ | $1.00 \pm 10^{-3}$ | $0.20 \pm 10^{-3}$ |
| | 1 | $\mathbf{1.40 \pm 10^{-3}}$ | $0.00 \pm 10^{-3}$ | $\mathbf{1.00 \pm 10^{-3}}$ | $\mathbf{0.20 \pm 10^{-3}}$ |
| 1 | 2000 | $1.00 \pm 10^{-3}$ | $\mathbf{0.42 \pm 10^{-3}}$ | $0.92 \pm 0.01$ | $1.00 \pm 10^{-3}$ |
| | 1000 | $0.61 \pm 0.16$ | $0.03 \pm 0.01$ | $0.97 \pm 0.01$ | $0.90 \pm 0.06$ |
| | 500 | $1.14 \pm 0.04$ | $0.00 \pm 10^{-3}$ | $1.00 \pm 10^{-3}$ | $0.41 \pm 0.04$ |
| | 100 | $1.35 \pm 0.01$ | $0.00 \pm 10^{-3}$ | $1.00 \pm 10^{-3}$ | $0.27 \pm 10^{-3}$ |
| | 10 | $1.40 \pm 10^{-3}$ | $0.00 \pm 10^{-3}$ | $1.00 \pm 10^{-3}$ | $0.20 \pm 10^{-3}$ |
| | 1 | $\mathbf{1.40 \pm 10^{-3}}$ | $0.00 \pm 10^{-3}$ | $\mathbf{1.00 \pm 10^{-3}}$ | $\mathbf{0.20 \pm 10^{-3}}$ |
| 3 | 2000 | $0.92 \pm 0.07$ | $\mathbf{0.02 \pm 0.01}$ | $0.99 \pm 10^{-3}$ | $0.91 \pm 0.05$ |
| | 1000 | $0.65 \pm 0.18$ | $0.00 \pm 10^{-3}$ | $1.00 \pm 10^{-3}$ | $0.69 \pm 0.14$ |
| | 500 | $1.07 \pm 0.07$ | $0.00 \pm 10^{-3}$ | $1.00 \pm 10^{-3}$ | $0.48 \pm 0.11$ |
| | 100 | $1.36 \pm 0.01$ | $0.00 \pm 10^{-3}$ | $1.00 \pm 10^{-3}$ | $0.27 \pm 0.01$ |
| | 10 | $1.40 \pm 10^{-3}$ | $0.00 \pm 10^{-3}$ | $1.00 \pm 10^{-3}$ | $0.20 \pm 10^{-3}$ |
| | 1 | $\mathbf{1.40 \pm 10^{-3}}$ | $0.00 \pm 10^{-3}$ | $\mathbf{1.00 \pm 10^{-3}}$ | $\mathbf{0.20 \pm 10^{-3}}$ |

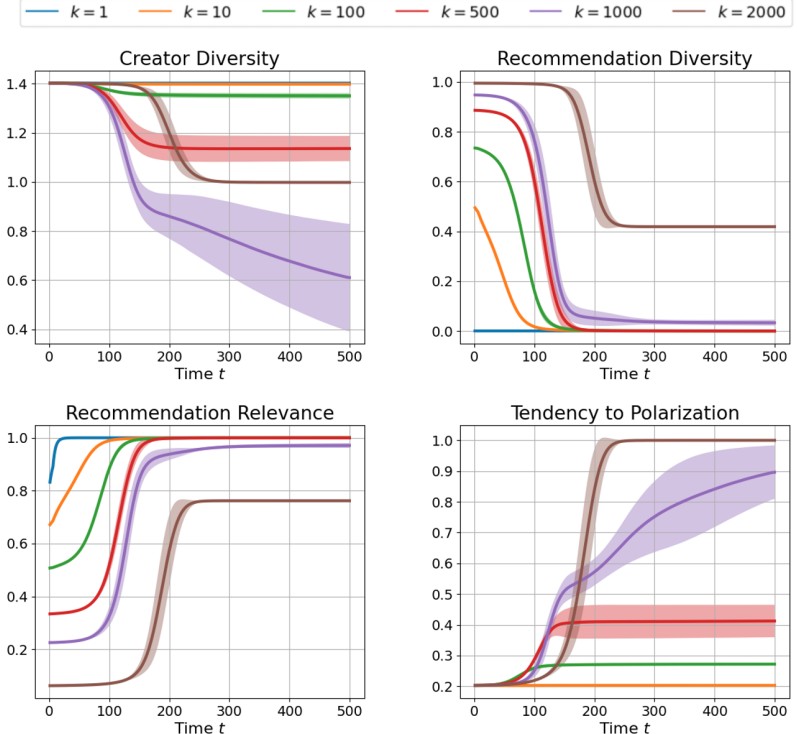

Figure 10: Changes of measures over time under different $k$, with $\beta = 1$, on MovieLens 20M dataset

Results for threshold truncation are in Table 5 and Figure 11. Similar to synthetic data, we see that a large (but not too large) threshold like 0.707 is good for improving CD and TP.

Table 5: Threshold truncation with different thresholds on MovieLens 20M dataset

| $\beta$ | threshold $\tau$ | CD | RD | RR | TP |
|---|---|---|---|---|---|
| | $-\cos(60°) = -0.5$ | $1.00 \pm 10^{-3}$ | $0.00 \pm 10^{-3}$ | $1.00 \pm 10^{-3}$ | $1.00 \pm 10^{-3}$ |
| | $-\cos(72°) = -0.309$ | $1.00 \pm 10^{-3}$ | $0.00 \pm 10^{-3}$ | $1.00 \pm 10^{-3}$ | $1.00 \pm 10^{-3}$ |
| | $\cos(90°) = 0$ | $0.01 \pm 0.01$ | $0.00 \pm 10^{-3}$ | $1.00 \pm 10^{-3}$ | $1.00 \pm 10^{-3}$ |
| 0 | $\cos(72°) = 0.309$ | $0.83 \pm 0.08$ | $0.00 \pm 10^{-3}$ | $1.00 \pm 10^{-3}$ | $0.72 \pm 0.09$ |
| | $\cos(60°) = 0.5$ | $1.20 \pm 0.05$ | $0.00 \pm 10^{-3}$ | $1.00 \pm 10^{-3}$ | $0.46 \pm 0.07$ |
| | $\cos(45°) = 0.707$ | $\mathbf{1.39 \pm 10^{-3}}$ | $0.00 \pm 10^{-3}$ | $1.00 \pm 10^{-3}$ | $\mathbf{0.20 \pm 10^{-3}}$ |
| | $\cos(30°) = 0.866$ | $1.36 \pm 10^{-3}$ | $0.00 \pm 10^{-3}$ | $1.00 \pm 10^{-3}$ | $0.40 \pm 0.01$ |
| | $-\cos(60°) = -0.5$ | $1.00 \pm 10^{-3}$ | $0.00 \pm 10^{-3}$ | $1.00 \pm 10^{-3}$ | $1.00 \pm 10^{-3}$ |
| | $-\cos(72°) = -0.309$ | $0.96 \pm 0.03$ | $0.00 \pm 10^{-3}$ | $1.00 \pm 10^{-3}$ | $0.95 \pm 0.03$ |
| | $\cos(90°) = 0$ | $0.02 \pm 0.02$ | $0.00 \pm 10^{-3}$ | $1.00 \pm 10^{-3}$ | $0.99 \pm 10^{-3}$ |
| 1 | $\cos(72°) = 0.309$ | $0.83 \pm 0.07$ | $0.00 \pm 10^{-3}$ | $1.00 \pm 10^{-3}$ | $0.66 \pm 0.10$ |
| | $\cos(60°) = 0.5$ | $1.18 \pm 0.06$ | $0.00 \pm 10^{-3}$ | $1.00 \pm 0.01$ | $0.44 \pm 0.07$ |
| | $\cos(45°) = 0.707$ | $\mathbf{1.40 \pm 10^{-3}}$ | $0.00 \pm 10^{-3}$ | $1.00 \pm 10^{-3}$ | $\mathbf{0.20 \pm 10^{-3}}$ |
| | $\cos(30°) = 0.866$ | $1.35 \pm 0.01$ | $0.00 \pm 10^{-3}$ | $1.00 \pm 10^{-3}$ | $0.40 \pm 0.02$ |
| | $-\cos(60°) = -0.5$ | $0.77 \pm 0.27$ | $0.00 \pm 10^{-3}$ | $1.00 \pm 10^{-3}$ | $0.86 \pm 0.09$ |
| | $-\cos(72°) = -0.309$ | $0.80 \pm 0.24$ | $0.00 \pm 10^{-3}$ | $1.00 \pm 10^{-3}$ | $0.79 \pm 0.13$ |
| | $\cos(90°) = 0$ | $0.04 \pm 0.02$ | $0.00 \pm 10^{-3}$ | $1.00 \pm 10^{-3}$ | $0.98 \pm 0.01$ |
| 3 | $\cos(72°) = 0.309$ | $0.99 \pm 0.11$ | $0.00 \pm 10^{-3}$ | $1.00 \pm 10^{-3}$ | $0.55 \pm 0.13$ |
| | $\cos(60°) = 0.5$ | $1.26 \pm 0.05$ | $0.00 \pm 10^{-3}$ | $1.00 \pm 10^{-3}$ | $0.36 \pm 0.06$ |
| | $\cos(45°) = 0.707$ | $\mathbf{1.40 \pm 10^{-3}}$ | $0.00 \pm 10^{-3}$ | $1.00 \pm 10^{-3}$ | $\mathbf{0.20 \pm 10^{-3}}$ |
| | $\cos(30°) = 0.866$ | $1.36 \pm 10^{-3}$ | $0.00 \pm 10^{-3}$ | $1.00 \pm 10^{-3}$ | $0.39 \pm 0.01$ |

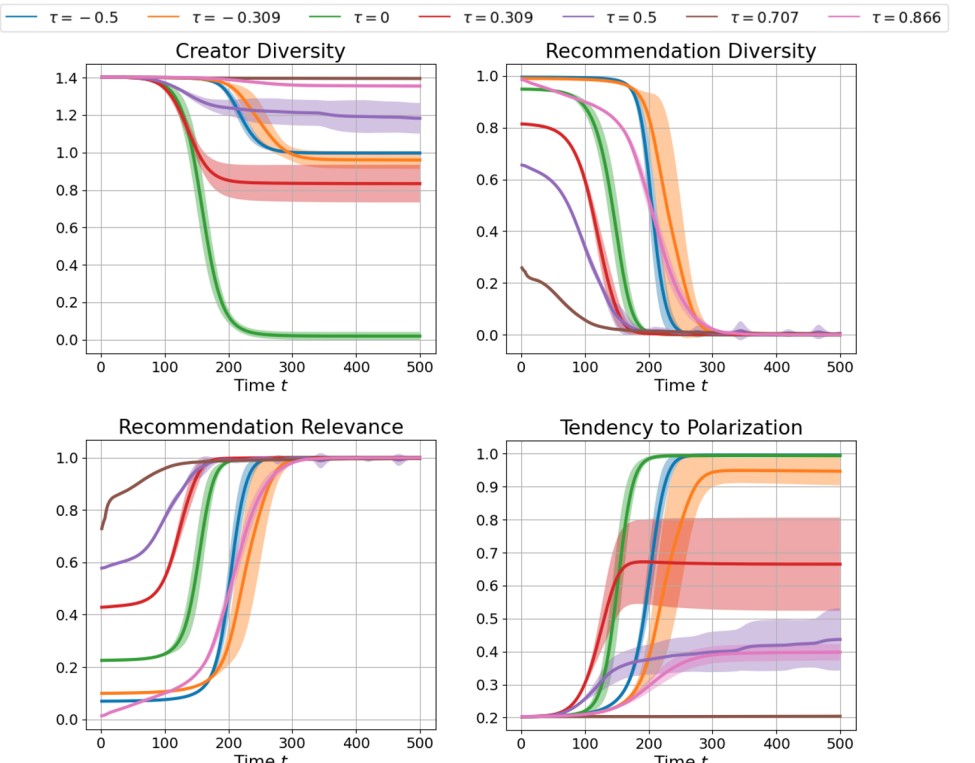

Figure 11: Changes of measures over time under truncation with different threshold $\tau$, with $\beta = 1$, on MovieLens 20M dataset

# D  Useful Lemmas

This section provides some lemmas that will be used in the proofs. They are mainly about some properties of the dynamics update rule.

**Claim D.1.** *For vectors $\boldsymbol{x}, \boldsymbol{y} \in \mathbb{R}^d$ with unit norm $\|\boldsymbol{x}\|_2 = \|\boldsymbol{y}\|_2 = 1$, we have:*

- $\|\boldsymbol{x} - \boldsymbol{y}\|_2^2 = 2(1 - \langle \boldsymbol{x}, \boldsymbol{y} \rangle)$.
- $\langle \boldsymbol{x}, \boldsymbol{y} \rangle = 1 - \frac{1}{2}\|\boldsymbol{x} - \boldsymbol{y}\|_2^2$.

**Lemma D.2** (Convex Cone Property)**.** *Let $\boldsymbol{z}_1, \ldots, \boldsymbol{z}_k \in \mathbb{R}^d$ be vectors with norm $\|\boldsymbol{z}_i^t\|_2 = 1$. Suppose $\langle \boldsymbol{z}_i, \boldsymbol{y} \rangle > 0$ for every $i = 1, \ldots, k$ for some $\boldsymbol{y} \in \mathbb{R}^d$. Let $\boldsymbol{x} = \mathcal{P}(\sum_{i=1}^{k} a_i \boldsymbol{z}_i)$ for some $a_1, \ldots, a_k \geq 0$ (namely, $\boldsymbol{x}$ is the normalization of some vector in the convex cone formed by $\boldsymbol{z}_1, \ldots, \boldsymbol{z}_k$). Then, we have*

$$\langle \boldsymbol{x}, \boldsymbol{y} \rangle \geq \min_{i=1}^{k} \langle \boldsymbol{z}_i, \boldsymbol{y} \rangle > 0 \quad and \quad \|\boldsymbol{x} - \boldsymbol{y}\|_2 \leq \max_{i=1}^{k} \|\boldsymbol{z}_i - \boldsymbol{y}\|_2 > 0.$$

*Proof.*

$$
\begin{aligned}
\langle \boldsymbol{x}, \boldsymbol{y} \rangle &= \Big\langle \frac{\sum_{i=1}^{k} a_i \boldsymbol{z}_i}{\|\sum_{i=1}^{k} a_i \boldsymbol{z}_i\|_2}, \boldsymbol{y} \Big\rangle = \frac{1}{\|\sum_{i=1}^{k} a_i \boldsymbol{z}_i\|_2} \sum_{i=1}^{k} a_i \langle \boldsymbol{z}_i, \boldsymbol{y} \rangle \\
&\geq \frac{1}{\|\sum_{i=1}^{k} a_i \boldsymbol{z}_i\|_2} \sum_{i=1}^{k} a_i \min_{i=1}^{k} \langle \boldsymbol{z}_i, \boldsymbol{y} \rangle = \min_{i=1}^{k} \langle \boldsymbol{z}_i, \boldsymbol{y} \rangle \frac{\sum_{i=1}^{k} a_i}{\|\sum_{i=1}^{k} a_i \boldsymbol{z}_i\|_2} \\
&\geq \min_{i=1}^{k} \langle \boldsymbol{z}_i, \boldsymbol{y} \rangle \frac{\sum_{i=1}^{k} a_i}{\sum_{i=1}^{k} a_i} = \min_{i=1}^{k} \langle \boldsymbol{z}_i, \boldsymbol{y} \rangle.
\end{aligned}
$$

This proves the first inequality. To prove the second inequality, we use Claim D.1 and the first inequality:

$$\|\boldsymbol{x} - \boldsymbol{y}\|_2 = \sqrt{2(1 - \langle \boldsymbol{x}, \boldsymbol{y} \rangle)} \leq \sqrt{2(1 - \min_i \langle \boldsymbol{z}_i, \boldsymbol{y} \rangle)} = \max_i \sqrt{2(1 - \langle \boldsymbol{z}_i, \boldsymbol{y} \rangle)} = \max_{i=1}^{k} \|\boldsymbol{z}_i - \boldsymbol{y}\|_2.$$

$\square$

**Lemma D.3.** *Let $\boldsymbol{x}^t, \boldsymbol{y}, \boldsymbol{z}^t \in \mathbb{R}^d$ be vectors with norm $\|\boldsymbol{x}^t\|_2 = 1$, $\|\boldsymbol{y}\|_2 \geq 0$, $\|\boldsymbol{z}^t\|_2 \leq 1$. Suppose $\langle \boldsymbol{x}^t, \boldsymbol{y} \rangle \geq 0$, $\langle \boldsymbol{z}^t, \boldsymbol{y} \rangle \geq 0$. After the update $\boldsymbol{x}^{t+1} = \mathcal{P}(\boldsymbol{x}^t + \eta \boldsymbol{z}^t)$, we have*

$$\langle \boldsymbol{x}^{t+1} - \boldsymbol{x}^t, \boldsymbol{y} \rangle \geq \frac{\eta}{1 + \eta \|\boldsymbol{z}^t\|_2} \Big( \langle \boldsymbol{z}^t, \boldsymbol{y} \rangle - \|\boldsymbol{z}^t\|_2 \langle \boldsymbol{x}^t, \boldsymbol{y} \rangle \Big).$$

*As a corollary, if $\boldsymbol{y} = \boldsymbol{z}^t$ and $\|\boldsymbol{z}^t\|_2 = 1$, then*

$$\langle \boldsymbol{x}^{t+1} - \boldsymbol{x}^t, \boldsymbol{z}^t \rangle \geq \frac{\eta}{1 + \eta} \Big( 1 - \langle \boldsymbol{x}^t, \boldsymbol{z}^t \rangle \Big).$$

*Proof.* By definition,

$$
\begin{aligned}
\langle \boldsymbol{x}^{t+1} - \boldsymbol{x}^t, \boldsymbol{y} \rangle &= \Big\langle \frac{\boldsymbol{x}^t + \eta \boldsymbol{z}^t}{\|\boldsymbol{x}^t + \eta \boldsymbol{z}^t\|_2} - \boldsymbol{x}^t, \boldsymbol{y} \Big\rangle \\
&= \Big( \frac{1}{\|\boldsymbol{x}^t + \eta \boldsymbol{z}^t\|_2} - 1 \Big) \cdot \langle \boldsymbol{x}^t, \boldsymbol{y} \rangle + \frac{\eta}{\|\boldsymbol{x}^t + \eta \boldsymbol{z}^t\|_2} \cdot \langle \boldsymbol{z}^t, \boldsymbol{y} \rangle \\
\text{(because } \|\boldsymbol{x}^t + \eta \boldsymbol{z}^t\|_2 \leq 1 + \eta \|\boldsymbol{z}^t\|_2) \quad &\geq \Big( \frac{1}{1 + \eta \|\boldsymbol{z}^t\|_2} - 1 \Big) \cdot \langle \boldsymbol{x}^t, \boldsymbol{y} \rangle + \frac{\eta}{1 + \eta \|\boldsymbol{z}^t\|_2} \cdot \langle \boldsymbol{z}^t, \boldsymbol{y} \rangle \\
&= \frac{\eta}{1 + \eta \|\boldsymbol{z}^t\|_2} \Big( \langle \boldsymbol{z}^t, \boldsymbol{y} \rangle - \|\boldsymbol{z}^t\|_2 \langle \boldsymbol{x}^t, \boldsymbol{y} \rangle \Big).
\end{aligned}
$$

$\square$

**Lemma D.4.** *Let $\boldsymbol{x}^t, \boldsymbol{z}^t \in \mathbb{R}^d$ be vectors with norm $\|\boldsymbol{x}^t\|_2 = 1$, $\|\boldsymbol{z}^t\|_2 \leq 1$. Suppose $\langle \boldsymbol{x}^t, \boldsymbol{z}^t \rangle \geq 0$ and $\eta > 0$. Then the update $\boldsymbol{x}^{t+1} = \mathcal{P}(\boldsymbol{x}^t + \eta \boldsymbol{z}^t)$ satisfies*

- $\langle \boldsymbol{x}^{t+1} - \boldsymbol{x}^t, \boldsymbol{z}^t \rangle \geq \frac{1}{\eta} \|\boldsymbol{x}^{t+1} - \boldsymbol{x}^t\|_2^2$.

- $\|\boldsymbol{x}^{t+1} - \boldsymbol{x}^t\|_2 \leq \eta \|\boldsymbol{z}^t\|_2$.

*Proof.* Let $\tilde{\boldsymbol{x}}^{t+1} = \boldsymbol{x}^t + \eta \boldsymbol{z}^t$, so $\boldsymbol{x}^t = \mathcal{P}(\tilde{\boldsymbol{x}}^{t+1})$ and $\boldsymbol{z}^t = \frac{1}{\eta}(\tilde{\boldsymbol{x}}^{t+1} - \boldsymbol{x}^t)$. Then we have

$$\langle \boldsymbol{x}^{t+1} - \boldsymbol{x}^t, \, \boldsymbol{z}^t \rangle \; = \; \frac{1}{\eta} \langle \boldsymbol{x}^{t+1} - \boldsymbol{x}^t, \, \tilde{\boldsymbol{x}}^{t+1} - \boldsymbol{x}^t \rangle.$$

Because $\langle \boldsymbol{x}^t, \boldsymbol{z}^t \rangle \geq 0$, the vector $\tilde{\boldsymbol{x}}^{t+1} = \boldsymbol{x}^t + \eta \boldsymbol{z}^t$ has length $\geq 1$ and hence is outside (or on the surface) of the $d$-dimensional unit ball. Since $\boldsymbol{x}^t = \mathcal{P}(\tilde{\boldsymbol{x}}^{t+1})$ is the projection of $\tilde{\boldsymbol{x}}^{t+1}$ onto the unit ball, and $\boldsymbol{z}^t$ is another vector inside the unit ball, by the "Pythagorean property" (Proposition 2.2 in [6]), we must have $\langle \boldsymbol{x}^t - \boldsymbol{x}^{t+1}, \tilde{\boldsymbol{x}}^{t+1} - \boldsymbol{x}^{t+1} \rangle \leq 0$. This implies

$$\langle \boldsymbol{x}^{t+1} - \boldsymbol{x}^t, \, \boldsymbol{z}^t \rangle \; \geq \; \frac{1}{\eta} \Big( \langle \boldsymbol{x}^{t+1} - \boldsymbol{x}^t, \, \tilde{\boldsymbol{x}}^{t+1} - \boldsymbol{x}^t \rangle + \langle \boldsymbol{x}^t - \boldsymbol{x}^{t+1}, \, \tilde{\boldsymbol{x}}^{t+1} - \boldsymbol{x}^{t+1} \rangle \Big)$$

$$= \; \frac{1}{\eta} \langle \boldsymbol{x}^{t+1} - \boldsymbol{x}^t, \, \boldsymbol{x}^{t+1} - \boldsymbol{x}^t \rangle \; = \; \frac{1}{\eta} \|\boldsymbol{x}^{t+1} - \boldsymbol{x}^t\|_2^2,$$

which proves the first claim. To prove the second claim, we use Cauchy-Schwarz inequality:

$$\frac{1}{\eta} \|\boldsymbol{x}^{t+1} - \boldsymbol{x}^t\|_2^2 \; \leq \; \langle \boldsymbol{x}^{t+1} - \boldsymbol{x}^t, \, \boldsymbol{z}^t \rangle \; \leq \; \|\boldsymbol{x}^{t+1} - \boldsymbol{x}^t\|_2 \|\boldsymbol{z}^t\|_2.$$

This implies $\|\boldsymbol{x}^{t+1} - \boldsymbol{x}^t\|_2 \leq \eta \|\boldsymbol{z}^t\|_2$. $\qquad \square$

**Lemma D.5.** *Consider a creator $\boldsymbol{v}_i^t$ and a user $\boldsymbol{u}_j^t$. Suppose the user is always recommended creator $i$ (so the user is updated by $\boldsymbol{u}_j^{t+1} = \mathcal{P}(\boldsymbol{u}_j^t + \eta_u f(\boldsymbol{v}_i^t, \boldsymbol{u}_j^t) \boldsymbol{v}_i^t))$, and creator $i$ is updated by $\boldsymbol{v}_i^{t+1} = \mathcal{P}(\boldsymbol{v}_i^t + \eta_c \boldsymbol{\alpha}_i^t)$ with $\|\boldsymbol{\alpha}_i^t\|_2 \leq 1$ and $\langle \boldsymbol{v}_i^t, \boldsymbol{\alpha}_i^t \rangle \geq 0$ at each time step. Assume:*

- *The inner product $\langle \boldsymbol{u}_j^0, \boldsymbol{v}_i^0 \rangle > 0$ initially. (Note that $\langle \boldsymbol{u}_j^0, \boldsymbol{u}_{j'}^0 \rangle$ needs not hold.)*

- *There exists some constant $L_f > 0$ such that $f(\boldsymbol{v}_i, \boldsymbol{u}_j) \geq L_f > 0$ whenever $\langle \boldsymbol{u}_j, \boldsymbol{v}_i \rangle > 0$.*

- *$\eta_c \leq \frac{\eta_u L_f}{2}$ and $0 \leq \eta_u < \frac{1}{2}$.*

*Then, we have $\langle \boldsymbol{u}_j^t, \boldsymbol{v}_i^t \rangle > 0$ in all time steps.*

*Proof.* We prove by induction. Suppose $\langle \boldsymbol{u}_j^t, \boldsymbol{v}_i^t \rangle > 0$ already holds. We prove that $\langle \boldsymbol{u}_j^{t+1}, \boldsymbol{v}_i^{t+1} \rangle > 0$ will also hold. Take the difference between $\langle \boldsymbol{u}_j^{t+1}, \boldsymbol{v}_i^{t+1} \rangle$ and $\langle \boldsymbol{u}_j^t, \boldsymbol{v}_i^t \rangle$:

$$\langle \boldsymbol{u}_j^{t+1}, \boldsymbol{v}_i^{t+1} \rangle - \langle \boldsymbol{u}_j^t, \boldsymbol{v}_i^t \rangle \; = \; \langle \boldsymbol{u}_j^{t+1}, \boldsymbol{v}_i^{t+1} - \boldsymbol{v}_i^t \rangle + \langle \boldsymbol{u}_j^{t+1} - \boldsymbol{u}_j^t, \boldsymbol{v}_i^t \rangle.$$

For $\langle \boldsymbol{u}_j^{t+1} - \boldsymbol{u}_j^t, \boldsymbol{v}_i^t \rangle$, using Lemma D.3 with $\boldsymbol{x}^t = \boldsymbol{u}_j^t$, $\boldsymbol{z}^t = \boldsymbol{v}_i^t$, and $\eta = \eta_u f(\boldsymbol{v}_i^t, \boldsymbol{u}_j^t)$, we get

$$\langle \boldsymbol{u}_j^{t+1} - \boldsymbol{u}_j^t, \boldsymbol{v}_i^t \rangle \; \geq \; \frac{\eta_u f(\boldsymbol{v}_i^t, \boldsymbol{u}_j^t)}{1 + \eta_u f(\boldsymbol{v}_i^t, \boldsymbol{u}_j^t)} \big( 1 - \langle \boldsymbol{u}_j^t, \boldsymbol{v}_i^t \rangle \big) \; \geq \; \frac{\eta_u L_f}{1 + \eta_u L_f} \big( 1 - \langle \boldsymbol{u}_j^t, \boldsymbol{v}_i^t \rangle \big).$$

For $\langle \boldsymbol{u}_j^{t+1}, \boldsymbol{v}_i^{t+1} - \boldsymbol{v}_i^t \rangle$, by Cauchy-Schwarz inequality and Lemma D.4,

$$\langle \boldsymbol{u}_j^{t+1}, \boldsymbol{v}_i^{t+1} - \boldsymbol{v}_i^t \rangle \; \geq \; - \|\boldsymbol{u}_j^{t+1}\|_2 \cdot \|\boldsymbol{v}_i^{t+1} - \boldsymbol{v}_i^t\|_2 \; \geq \; -1 \cdot \eta_c \|\boldsymbol{\alpha}_i^t\|_2 \; \geq \; -\eta_c.$$

- If $1 - \langle \boldsymbol{u}_j^t, \boldsymbol{v}_i^t \rangle > \frac{1}{2}(1 + \eta_u L_f)$, then we have

$$\langle \boldsymbol{u}_j^{t+1}, \boldsymbol{v}_i^{t+1} \rangle - \langle \boldsymbol{u}_j^t, \boldsymbol{v}_i^t \rangle \; > \; \eta_u L_f \tfrac{1}{2} - \eta_c \; \geq 0$$

  by the assumption of $\eta_c \leq \frac{\eta_u L_f}{2}$.

- If $1 - \langle \boldsymbol{u}_j^t, \boldsymbol{v}_i^t \rangle \leq \frac{1}{2}(1 + \eta_u L_f)$, then we have

$$\langle \boldsymbol{u}_j^{t+1}, \boldsymbol{v}_i^{t+1} \rangle - \langle \boldsymbol{u}_j^t, \boldsymbol{v}_i^t \rangle \; \geq \; 0 - \eta_c$$

$$\implies \langle \boldsymbol{u}_j^{t+1}, \boldsymbol{v}_i^{t+1} \rangle \; \geq \; \langle \boldsymbol{u}_j^t, \boldsymbol{v}_i^t \rangle - \eta_c \; \geq \; \tfrac{1}{2} - \tfrac{1}{2} \eta_u L_f - \eta_c \; > \; 0$$

  under the assumption of $\eta_c \leq \frac{\eta_u L_f}{2}$ and $\eta_u < \frac{1}{2}$.

The above two cases together ensure $\langle \boldsymbol{u}_j^{t+1}, \boldsymbol{v}_i^{t+1} \rangle > 0$.  □

**Lemma D.6.** *Consider a system of one user and one creator that satisfies $\langle \boldsymbol{u}_j^0, \boldsymbol{v}_i^0 \rangle > 0$ and $\langle \boldsymbol{u}_j^0, \boldsymbol{y} \rangle > \langle \boldsymbol{v}_i^0, \boldsymbol{y} \rangle > 0$ for some $\boldsymbol{y} \in \mathbb{R}^d$ with $\|\boldsymbol{y}\| \leq 1$ initially. The creator is always recommended to the user (so the updates are $\boldsymbol{u}_j^{t+1} = \mathcal{P}(\boldsymbol{u}_j^t + \eta_u f(\boldsymbol{v}_i^t, \boldsymbol{u}_j^t) \boldsymbol{v}_i^t)$ and $\boldsymbol{v}_i^{t+1} = \mathcal{P}(\boldsymbol{v}_i^t + \eta_c \boldsymbol{u}_j^t))$. Suppose $\eta_c \leq \frac{\eta_u L_f}{2}$ and $0 \leq \eta_u < \frac{1}{2}$. Then, we have:*

- $\langle \boldsymbol{u}_j^t, \boldsymbol{y} \rangle > \langle \boldsymbol{v}_i^t, \boldsymbol{y} \rangle > 0$ for all $t \geq 1$.

- *Suppose $\langle \boldsymbol{u}_j^0, \boldsymbol{y} \rangle - \langle \boldsymbol{v}_i^0, \boldsymbol{y} \rangle = D > 0$. For any $R < D$, after $T = \frac{8}{3\eta_u L_f} \ln \frac{2}{R^2}$ steps, we have $\langle \boldsymbol{v}_i^T, \boldsymbol{y} \rangle - \langle \boldsymbol{v}_i^0, \boldsymbol{y} \rangle \geq \frac{\eta_c}{\eta_u + \eta_c}(D - R)$.*

*Proof.* We prove the first item by induction. Suppose $\langle \boldsymbol{u}_j^t, \boldsymbol{y} \rangle > \langle \boldsymbol{v}_i^t, \boldsymbol{y} \rangle > 0$ holds. Consider $t + 1$. First, by Lemma D.2, $\langle \boldsymbol{v}_i^{t+1}, \boldsymbol{y} \rangle > 0$ holds. Then, we prove $\langle \boldsymbol{u}_j^{t+1}, \boldsymbol{y} \rangle > \langle \boldsymbol{v}_i^{t+1}, \boldsymbol{y} \rangle$. Let $f = f(\boldsymbol{v}_i^t, \boldsymbol{u}_j^t)$.

$$
\begin{aligned}
\langle \boldsymbol{u}_j^{t+1}, \boldsymbol{y} \rangle - \langle \boldsymbol{v}_i^{t+1}, \boldsymbol{y} \rangle &= \left\langle \frac{\boldsymbol{u}_j^t + \eta_u f \boldsymbol{v}_i^t}{\|\boldsymbol{u}_j^t + \eta_u f \boldsymbol{v}_i^t\|_2}, \boldsymbol{y} \right\rangle - \left\langle \frac{\boldsymbol{v}_i^t + \eta_c \boldsymbol{u}_j^t}{\|\boldsymbol{v}_i^t + \eta_c \boldsymbol{u}_j^t\|_2}, \boldsymbol{y} \right\rangle \\
&= \left( \frac{1}{\|\boldsymbol{u}_j^t + \eta_u f \boldsymbol{v}_i^t\|_2} - \frac{\eta_c}{\|\boldsymbol{v}_i^t + \eta_c \boldsymbol{u}_j^t\|_2} \right) \langle \boldsymbol{u}_j^t, \boldsymbol{y} \rangle - \left( \frac{1}{\|\boldsymbol{v}_i^t + \eta_c \boldsymbol{u}_j^t\|_2} - \frac{\eta_u f}{\|\boldsymbol{u}_j^t + \eta_u f \boldsymbol{v}_i^t\|_2} \right) \langle \boldsymbol{v}_i^t, \boldsymbol{y} \rangle \\
&> \left( \frac{1}{\|\boldsymbol{u}_j^t + \eta_u f \boldsymbol{v}_i^t\|_2} - \frac{\eta_c}{\|\boldsymbol{v}_i^t + \eta_c \boldsymbol{u}_j^t\|_2} \right) \langle \boldsymbol{v}_i^t, \boldsymbol{y} \rangle - \left( \frac{1}{\|\boldsymbol{v}_i^t + \eta_c \boldsymbol{u}_j^t\|_2} - \frac{\eta_u f}{\|\boldsymbol{u}_j^t + \eta_u f \boldsymbol{v}_i^t\|_2} \right) \langle \boldsymbol{v}_i^t, \boldsymbol{y} \rangle \\
&= \left( \frac{1 + \eta_u f}{\|\boldsymbol{u}_j^t + \eta_u f \boldsymbol{v}_i^t\|_2} - \frac{1 + \eta_c}{\|\boldsymbol{v}_i^t + \eta_c \boldsymbol{u}_j^t\|_2} \right) \langle \boldsymbol{v}_i^t, \boldsymbol{y} \rangle \\
&= \left( \frac{1 + \eta_u f}{\sqrt{1 + 2\eta_u f \langle \boldsymbol{u}_j^t, \boldsymbol{v}_i^t \rangle + (\eta_u f)^2}} - \frac{1 + \eta_c}{\sqrt{1 + 2\eta_c \langle \boldsymbol{u}_j^t, \boldsymbol{v}_i^t \rangle + (\eta_c)^2}} \right) \langle \boldsymbol{v}_i^t, \boldsymbol{y} \rangle.
\end{aligned}
$$

Let $a = \langle \boldsymbol{u}_j^t, \boldsymbol{v}_i^t \rangle \leq 1$. We note that the function

$$
h(\eta) = \frac{1 + \eta}{\sqrt{1 + 2\eta a + \eta^2}} = \sqrt{\frac{1 + 2\eta + \eta^2}{1 + 2\eta a + \eta^2}} = \sqrt{1 + \frac{(2 - 2a)\eta}{1 + 2\eta a + \eta^2}} = \sqrt{1 + \frac{2(1 - a)}{\frac{1}{\eta} + 2a + \eta}}
$$

is increasing in $\eta \in [0, 1]$. Under the assumption of $\eta_c \leq \frac{\eta_u L_f}{2} \leq \frac{\eta_u f}{2} < \eta_u f$, we have $h(\eta_c) \leq h(\eta_u f)$ and hence

$$
\langle \boldsymbol{u}_j^{t+1}, \boldsymbol{y} \rangle - \langle \boldsymbol{v}_i^{t+1}, \boldsymbol{y} \rangle > \left( h(\eta_u f) - h(\eta_c) \right) \langle \boldsymbol{v}_i^t, \boldsymbol{y} \rangle \geq 0.
$$

We then prove the second item. Using Lemma D.3 for $\boldsymbol{v}_i^{t+1} = \mathcal{P}(\boldsymbol{v}_i^t + \eta_c \boldsymbol{u}_j^t)$, we get

$$
\langle \boldsymbol{v}_i^{t+1} - \boldsymbol{v}_i^t, \boldsymbol{y} \rangle \geq \frac{\eta_c}{1 + \eta_c} \left( \langle \boldsymbol{u}_j^t, \boldsymbol{y} \rangle - \langle \boldsymbol{v}_i^t, \boldsymbol{y} \rangle \right).
$$

Using Lemma D.3 for $\boldsymbol{u}_j^{t+1} = \mathcal{P}(\boldsymbol{u}_j^t + \eta_u f(\boldsymbol{v}_i^t, \boldsymbol{u}_j^t) \boldsymbol{v}_i^t)$ and using the fact $\langle \boldsymbol{v}_i^t, \boldsymbol{y} \rangle - \langle \boldsymbol{u}_j^t, \boldsymbol{y} \rangle < 0$ proved in item 1,

$$
\langle \boldsymbol{u}_j^{t+1} - \boldsymbol{u}_j^t, \boldsymbol{y} \rangle \geq \frac{\eta_u f(\boldsymbol{v}_i^t, \boldsymbol{u}_j^t)}{1 + \eta_u f(\boldsymbol{v}_i^t, \boldsymbol{u}_j^t)} \left( \langle \boldsymbol{v}_i^t, \boldsymbol{y} \rangle - \langle \boldsymbol{u}_j^t, \boldsymbol{y} \rangle \right) \geq \frac{\eta_u}{1 + \eta_u} \left( \langle \boldsymbol{v}_i^t, \boldsymbol{y} \rangle - \langle \boldsymbol{u}_j^t, \boldsymbol{y} \rangle \right).
$$

Rearranging the above two inequalities:

$$
\frac{1 + \eta_c}{\eta_c} \left( \langle \boldsymbol{v}_i^{t+1}, \boldsymbol{y} \rangle - \langle \boldsymbol{v}_i^t, \boldsymbol{y} \rangle \right) \geq \langle \boldsymbol{u}_j^t, \boldsymbol{y} \rangle - \langle \boldsymbol{v}_i^t, \boldsymbol{y} \rangle;
$$

$$
\frac{1 + \eta_u}{\eta_u} \left( \langle \boldsymbol{u}_j^{t+1}, \boldsymbol{y} \rangle - \langle \boldsymbol{u}_j^t, \boldsymbol{y} \rangle \right) \geq \langle \boldsymbol{v}_i^t, \boldsymbol{y} \rangle - \langle \boldsymbol{u}_j^t, \boldsymbol{y} \rangle.
$$

Summing the above two inequalities over $t = 0, 1, \ldots, T - 1$:

$$\frac{1 + \eta_c}{\eta_c}\left(\langle \boldsymbol{v}_i^T, \boldsymbol{y}\rangle - \langle \boldsymbol{v}_i^0, \boldsymbol{y}\rangle\right) + \frac{1 + \eta_u}{\eta_u}\left(\langle \boldsymbol{u}_j^T, \boldsymbol{y}\rangle - \langle \boldsymbol{u}_j^0, \boldsymbol{y}\rangle\right) \geq 0. \tag{6}$$

According to Lemma F.1, after at most $T = \frac{8}{3\eta_u L_f}\ln\frac{2}{R^2}$ steps, we have $\|\boldsymbol{u}_j^T - \boldsymbol{v}_i^T\|_2 \leq R$. This implies $\langle \boldsymbol{u}_j^T, \boldsymbol{y}\rangle - \langle \boldsymbol{v}_i^T, \boldsymbol{y}\rangle = \langle \boldsymbol{u}_j^T - \boldsymbol{v}_i^T, \boldsymbol{y}\rangle \leq \|\boldsymbol{u}_j^T - \boldsymbol{v}_i^T\| \leq R$ and hence

$$\left(\langle \boldsymbol{v}_i^T, \boldsymbol{y}\rangle - \langle \boldsymbol{v}_i^0, \boldsymbol{y}\rangle\right) - \left(\langle \boldsymbol{u}_j^T, \boldsymbol{y}\rangle - \langle \boldsymbol{u}_j^0, \boldsymbol{y}\rangle\right) = \left(\langle \boldsymbol{u}_j^0, \boldsymbol{y}\rangle - \langle \boldsymbol{v}_i^0, \boldsymbol{y}\rangle\right) - \left(\langle \boldsymbol{u}_j^T, \boldsymbol{y}\rangle - \langle \boldsymbol{v}_i^T, \boldsymbol{y}\rangle\right) \geq D - R. \tag{7}$$

Multiplying (7) by $\frac{1+\eta_u}{\eta_u}$ and adding to (6):

$$\left(\frac{1 + \eta_c}{\eta_c} + \frac{1 + \eta_u}{\eta_u}\right)\left(\langle \boldsymbol{v}_i^T, \boldsymbol{y}\rangle - \langle \boldsymbol{v}_i^0, \boldsymbol{y}\rangle\right) \geq \frac{1 + \eta_u}{\eta_u}(D - R).$$

This implies

$$\langle \boldsymbol{v}_i^T, \boldsymbol{y}\rangle - \langle \boldsymbol{v}_i^0, \boldsymbol{y}\rangle \geq \frac{\frac{1+\eta_u}{\eta_u}}{\frac{1+\eta_c}{\eta_c} + \frac{1+\eta_u}{\eta_u}}(D-R) = \frac{\eta_c(1 + \eta_u)}{\eta_u(1 + \eta_c) + \eta_c(1 + \eta_u)}(D-R) \geq \frac{\eta_c}{\eta_u + \eta_c}(D-R).$$

given $\eta_c \leq \eta_u$. $\qquad\square$

The following lemma shows that, when we *reflect* some of the feature vectors in a system $(U^t, V^t) = (\{\boldsymbol{u}_j^t\}_{j\in[m]}, \{\boldsymbol{v}_i^t\}_{i\in[n]})$, there is a correspondence between the behaviors of the system with the reflected vectors and the original system.

**Lemma D.7** (Reflection). *Let $(U^t, V^t) = (\{\boldsymbol{u}_j^t\}_{j\in[m]}, \{\boldsymbol{v}_i^t\}_{i\in[n]})$ be a system of $m$ users and $n$ creators with impact functions $f, g$. Let $a_i, b_j \in \{+1, -1\}$, $\forall i \in [n]$, $\forall i \in [m]$ be some binary constants. Define:*

$$\tilde{\boldsymbol{u}}_j^t = b_j \boldsymbol{u}_j^t = \pm\boldsymbol{u}_j^t, \qquad \tilde{\boldsymbol{v}}_i^t = a_i \boldsymbol{v}_i^t = \pm\tilde{\boldsymbol{v}}_i^t.$$

*and impact functions*

$$\tilde{f}(\tilde{\boldsymbol{v}}_i, \tilde{\boldsymbol{u}}_j) = a_i b_j f(\boldsymbol{v}_i, \boldsymbol{u}_j), \qquad \tilde{g}(\tilde{\boldsymbol{u}}_j, \tilde{\boldsymbol{v}}_i) = a_i b_j g(\boldsymbol{u}_j, \boldsymbol{v}_i).$$

*Then:*

- *There is a "correspondence" between the evolution of the system $(U^t, V^t)$ with impact functions $f, g$ and the evolution of the system $(\tilde{U}^t, \tilde{V}^t) = (\{\tilde{\boldsymbol{u}}_j^t\}_{j\in[m]}, \{\tilde{\boldsymbol{v}}_i^t\}_{i\in[n]})$ with impact functions $\tilde{f}, \tilde{g}$. Formally, suppose every user is recommended the same creator in the two systems, then the updated vectors in the two systems still satisfy the relations: $\tilde{\boldsymbol{u}}_j^{t+1} = b_j \boldsymbol{u}_j^{t+1}$, $\tilde{\boldsymbol{v}}_i^{t+1} = a_i \boldsymbol{v}_i^{t+1}$.*

- *If the system $(\tilde{U}^t, \tilde{V}^t)$ is in R-bi-polarization, then the original system $(U^t, V^t)$ is also in R-bi-polarization.*

*Proof.* Consider the first item. Suppose user $i$ is recommended creator $j$ at time step $t$ in the two systems. Then by definition, the updated user vectors in the two systems satisfy

$$\begin{aligned}
\tilde{\boldsymbol{u}}_j^{t+1} &= \mathcal{P}\left(\tilde{\boldsymbol{u}}_j^t + \eta_u \tilde{f}(\tilde{\boldsymbol{v}}_i^t, \tilde{\boldsymbol{u}}_j^t)\tilde{\boldsymbol{v}}_i^t\right) = \mathcal{P}\left(b_j \boldsymbol{u}_j^t + \eta_u a_i b_j f(\boldsymbol{v}_i^t, \boldsymbol{u}_j^t)a_i \boldsymbol{v}_i^t\right) \\
&= \mathcal{P}\left(b_j \boldsymbol{u}_j^t + \eta_u b_j f(\boldsymbol{v}_i^t, \boldsymbol{u}_j^t)\boldsymbol{v}_i^t\right) = b_j \mathcal{P}\left(\boldsymbol{u}_j^t + \eta_u f(\boldsymbol{v}_i^t, \boldsymbol{u}_j^t)\boldsymbol{v}_i^t\right) = b_j \boldsymbol{u}_j^{t+1}
\end{aligned}$$

Suppose creator $i$ is recommended to the set of users $J$ at time step $t$ in the two systems. Then,

$$\begin{aligned}
\tilde{\boldsymbol{v}}_i^{t+1} &= \mathcal{P}\left(\tilde{\boldsymbol{v}}_i^t + \frac{\eta_c}{|J|}\sum_{j\in J} g(\tilde{\boldsymbol{u}}_j^t, \tilde{\boldsymbol{v}}_i^t)\tilde{\boldsymbol{u}}_j^t\right) \\
&= \mathcal{P}\left(a_i \boldsymbol{v}_i^t + \frac{\eta_c}{|J|}\sum_{j\in J} a_i b_j g(\boldsymbol{u}_j^t, \boldsymbol{v}_i^t)b_j \boldsymbol{u}_j^t\right) \\
&= \mathcal{P}\left(a_i \boldsymbol{v}_i^t + \frac{\eta_c}{|J|}\sum_{j\in J} a_i g(\boldsymbol{u}_j^t, \boldsymbol{v}_i^t)\boldsymbol{u}_j^t\right) \\
&= a_i \mathcal{P}\left(\boldsymbol{v}_i^t + \frac{\eta_c}{|J|}\sum_{j\in J} g(\boldsymbol{u}_j^t, \boldsymbol{v}_i^t)\boldsymbol{u}_j^t\right) = a_i \boldsymbol{v}_i^{t+1}.
\end{aligned}$$

This means that the evolution of the system $(\tilde{U}^t, \tilde{V}^t)$ has a correspondence to the evolution of the original system $(U^t, V^t)$.

Consider the second item. Suppose $(\tilde{U}^t, \tilde{V}^t)$ is in $R$-bi-polarization, so $\tilde{v}_i^t = \pm v_i^t$ is $R$-close to $\pm c$ and $\tilde{u}_j^t = \pm u_j^t$ is $R$-close to $\pm c$ with some vector $c \in \mathbb{S}^{d-1}$. This implies that $v_i^t$ is $R$-close to $\pm c$ and $u_j^t$ is $R$-close to $\pm c$. So, the system $(U^t, V^t)$ satisfies $R$-bi-polarization. $\qquad\square$

## E   Proof of Proposition 3.2

*Proof.* Let $(\boldsymbol{U}^t, \boldsymbol{V}^t)$ be an $(R, \boldsymbol{c})$-bi-polarization state with $R \in [0, 1]$ and $\boldsymbol{c} \in \mathbb{S}^{d-1}$, where all $\boldsymbol{u}_j^t$ and $\boldsymbol{v}_i^t$ are within distance $R$ to $+\boldsymbol{c}$ or $-\boldsymbol{c}$. We show that, after one step of update, $\boldsymbol{u}_j^{t+1}$ and $\boldsymbol{v}_i^{t+1}$ are still within distance $R$ to $+\boldsymbol{c}$ or $-\boldsymbol{c}$, so $(\boldsymbol{U}^{t+1}, \boldsymbol{V}^{t+1})$ still satisfies $(R, \boldsymbol{c})$-bi-polarization.

Consider $\boldsymbol{u}_j^t$. Without loss of generality, suppose $\boldsymbol{u}_j^t$ is close to $+\boldsymbol{c}$, so $\|\boldsymbol{u}_j^t - \boldsymbol{c}\|_2 \le R$. Suppose user $j$ is recommended creator $i$ at step $t$. Let $\tilde{\boldsymbol{v}}_i^t = \boldsymbol{v}_i^t$ if $\langle \boldsymbol{v}_i^t, \boldsymbol{u}_j^t \rangle \ge 0$ and $\tilde{\boldsymbol{v}}_i^t = -\boldsymbol{v}_i^t$ if $\langle \boldsymbol{v}_i^t, \boldsymbol{u}_j^t \rangle < 0$. Then, the user update is

$$\boldsymbol{u}_j^{t+1} = \mathcal{P}\Big(\boldsymbol{u}_j^t + \eta_u f(\boldsymbol{v}_i^t, \boldsymbol{u}_j^t)\boldsymbol{v}_i^t\Big) = \mathcal{P}\Big(\boldsymbol{u}_j^t + \eta_u |f(\boldsymbol{v}_i^t, \boldsymbol{u}_j^t)|\tilde{\boldsymbol{v}}_i^t\Big).$$

Since $\tilde{\boldsymbol{v}}_i^t$ is close to $+\boldsymbol{c}$ or $-\boldsymbol{c}$, $\langle \tilde{\boldsymbol{v}}_i^t, \boldsymbol{u}_j^t \rangle > 0$, and $\boldsymbol{u}_j^t$ is close to $+\boldsymbol{c}$, it must be that $\tilde{\boldsymbol{v}}_i^t$ is close to $+\boldsymbol{c}$, so $\|\tilde{\boldsymbol{v}}_i^t - \boldsymbol{c}\|_2 \le R$. Then, since $\boldsymbol{u}_j^{t+1}$ is the normalization of a vector in the convex cone formed by $\boldsymbol{u}_j^t$ and $\tilde{\boldsymbol{v}}_i^t$, by Lemma D.2, we have

$$\|\boldsymbol{u}_j^{t+1} - \boldsymbol{c}\|_2 \ \le \ \max\big\{\|\boldsymbol{u}_j^t - \boldsymbol{c}\|_2, \|\tilde{\boldsymbol{v}}_i^t - \boldsymbol{c}\|_2\big\} \ \le \ R.$$

Consider $\boldsymbol{v}_i^t$. Suppose $\|\boldsymbol{v}_i^t - \boldsymbol{c}\|_2 \le R$. Let $J$ be the set of users that are recommended creator $i$ at step $t$. For each $j \in J$, let $\tilde{\boldsymbol{u}}_j^t = \boldsymbol{u}_j^t$ if $\langle \boldsymbol{u}_j^t, \boldsymbol{v}_i^t \rangle \ge 0$ and $\tilde{\boldsymbol{u}}_j^t = -\boldsymbol{u}_j^t$ if $\langle \boldsymbol{u}_j^t, \boldsymbol{v}_i^t \rangle < 0$. Then, the creator update is

$$\boldsymbol{v}_i^{t+1} = \mathcal{P}\Big(\boldsymbol{v}_i^t + \frac{\eta_c}{|J|}\sum_{j \in J} g(\boldsymbol{u}_j^t, \boldsymbol{v}_i^t)\boldsymbol{u}_j^t\Big) = \mathcal{P}\Big(\boldsymbol{v}_i^t + \frac{\eta_c}{|J|}\sum_{j \in J} |g(\boldsymbol{u}_j^t, \boldsymbol{v}_i^t)|\tilde{\boldsymbol{u}}_j^t\Big).$$

We note that every $\tilde{\boldsymbol{u}}_j^t$ satisfies $\|\tilde{\boldsymbol{u}}_j^t - \boldsymbol{c}\|_2 \le R$ (by the same reasoning as above). Then, since $\boldsymbol{v}_i^{t+1}$ is the normalization of a vector in the convex cone formed by $\boldsymbol{v}_i^t$ and $\{\tilde{\boldsymbol{u}}_j^t\}_{j \in J}$, by Lemma D.2, we have

$$\|\boldsymbol{v}_i^{t+1} - \boldsymbol{c}\|_2 \ \le \ \min\Big\{\|\boldsymbol{v}_i^t - \boldsymbol{c}\|_2, \ \min_{j \in J}\|\tilde{\boldsymbol{u}}_j^t - \boldsymbol{c}\|_2\Big\} \ \le \ R. \qquad\square$$

## F   Proof of Lemma 3.4

Lemma 3.4 is proved by induction on the number $n$ of creators. We first show that any system with 1 creator and multiple users must converge to $R$-bi-polarization in finite steps for any $R > 0$. Using the result for 1 creator, we then construct a finite length path that leads to $R$-bi-polarization for any system with $n \ge 2$ creators.

### F.1   Base Case: Convergence Results for $n = 1$ Creator

We prove some convergence results for the special case of only one creator. This will serve as the basis for the proof for $n \ge 2$ creators. Recall that we have the following dynamics update rule:

- User: $\boldsymbol{u}_j^{t+1} = \mathcal{P}(\boldsymbol{u}_j^t + \eta_u f(\boldsymbol{v}_i^t, \boldsymbol{u}_j^t)\boldsymbol{v}_i^t)$ where $\boldsymbol{v}_i^t$ is the creator recommended to user $j$; $f(\boldsymbol{v}_i, \boldsymbol{u}_j)$ satisfies:

$$f(\boldsymbol{v}_i, \boldsymbol{u}_j) \text{ is } \begin{cases} > 0 & \text{if } \langle \boldsymbol{v}_i, \boldsymbol{u}_j \rangle > 0 \\ < 0 & \text{if } \langle \boldsymbol{v}_i, \boldsymbol{u}_j \rangle < 0 \\ = 0 & \text{if } \langle \boldsymbol{v}_i, \boldsymbol{u}_j \rangle = 0. \end{cases} \tag{8}$$

- Creator: $\boldsymbol{v}_i^{t+1} = \mathcal{P}(\boldsymbol{v}_j^t + \frac{\eta_c}{|J|}\sum_{j \in J} g(\boldsymbol{u}_j^t, \boldsymbol{v}_i^t)\boldsymbol{u}_j^t)$ where $J$ is the set of users being recommended creator $i$.

**Lemma F.1.** *Consider a system of* $1$ *creator* $\boldsymbol{v}_i^t$ *and* $|J|$ *users* $\{\boldsymbol{u}_j^t\}_{j \in J}$, *where the creator is recommended to all users at every time step. Assume:*

- *Initially,* $\forall j \in J, \langle \boldsymbol{u}_j^0, \boldsymbol{v}_i^0 \rangle > 0$.

- *There exists some constant* $L_f > 0$ *such that* $f(\boldsymbol{v}_i, \boldsymbol{u}_j) \geq L_f > 0$ *whenever* $\langle \boldsymbol{v}_i, \boldsymbol{u}_j \rangle > 0$.

- $g(\boldsymbol{u}_j, \boldsymbol{v}_i) = 1$ *when* $\langle \boldsymbol{u}_j, \boldsymbol{v}_i \rangle > 0$.

- $\eta_c \leq \frac{\eta_u L_f}{2}$ *and* $0 \leq \eta_u < \frac{1}{2}$.

*Then, for any* $R > 0$, *after at most* $\frac{8}{3 \eta_u L_f} \ln \frac{2|J|}{R^2}$ *steps,* $\sum_{j \in J} \|\boldsymbol{u}_j^t - \boldsymbol{v}_i^t\|_2^2 \leq R^2$ *will hold forever. In particular, each user vector will satisfy* $\|\boldsymbol{u}_j^t - \boldsymbol{v}_i^t\|_2 \leq R$.

*Proof.* We first note that, by Lemma D.5, all user vectors satisfy $\langle \boldsymbol{u}_j^t, \boldsymbol{v}_i^t \rangle > 0$ in all time steps $t > 0$. Hence, the creator update is always $\boldsymbol{v}_i^{t+1} = \mathcal{P}(\boldsymbol{v}_i^t + \frac{\eta_c}{|J|} \sum_{j \in J} g(\boldsymbol{u}_j^t, \boldsymbol{v}_i^t) \boldsymbol{u}_j^t) = \mathcal{P}(\boldsymbol{v}_i^t + \eta_c \frac{1}{|J|} \sum_{j \in J} \boldsymbol{u}_j^t)$.

Let $a_t = 1/(1 - \frac{3 \eta_u L_f}{8})^t$. Define the following potential function:

$$\Phi^t = a_t \sum_{j \in J} \frac{1}{2} \|\boldsymbol{u}_j^t - \boldsymbol{v}_i^t\|_2^2 = a_t \sum_{j \in J} \left(1 - \langle \boldsymbol{u}_j^t, \boldsymbol{v}_i^t \rangle\right). \tag{9}$$

We will show that $\Phi^t$ is monotonically decreasing. Take the difference between $\Phi^{t+1}$ and $\Phi^t$:

$$\Phi^{t+1} - \Phi^t = a_{t+1} \sum_{j \in J} \left(\langle \boldsymbol{u}_j^t, \boldsymbol{v}_i^t \rangle - \langle \boldsymbol{u}_j^{t+1}, \boldsymbol{v}_i^{t+1} \rangle\right) + (a_{t+1} - a_t) \sum_{j \in J} \left(1 - \langle \boldsymbol{u}_j^t, \boldsymbol{v}_i^t \rangle\right)$$

$$= a_{t+1} \left(\sum_{j \in J} \langle \boldsymbol{v}_i^t, \boldsymbol{u}_j^t - \boldsymbol{u}_j^{t+1} \rangle + \sum_{j \in J} \langle \boldsymbol{u}_j^t, \boldsymbol{v}_i^t - \boldsymbol{v}_i^{t+1} \rangle + \sum_{j \in J} \langle \boldsymbol{u}_j^{t+1} - \boldsymbol{u}_j^t, \boldsymbol{v}_i^t - \boldsymbol{v}_i^{t+1} \rangle\right)$$

$$+ (a_{t+1} - a_t) \sum_{j \in J} \left(1 - \langle \boldsymbol{u}_j^t, \boldsymbol{v}_i^t \rangle\right).$$

Using Lemma D.3 with $\boldsymbol{x}^t = \boldsymbol{u}_j^t$, $\boldsymbol{z}^t = \boldsymbol{v}_i^t$, and $\eta = \eta_u f(\boldsymbol{v}_i^t, \boldsymbol{u}_j^t)$, we get

$$\langle \boldsymbol{v}_i^t, \boldsymbol{u}_j^t - \boldsymbol{u}_j^{t+1} \rangle \leq -\frac{\eta_u f(\boldsymbol{v}_i^t, \boldsymbol{u}_j^t)}{1 + \eta_u f(\boldsymbol{v}_i^t, \boldsymbol{u}_j^t)} \left(1 - \langle \boldsymbol{u}_j^t, \boldsymbol{v}_i^t \rangle\right) \leq -\frac{\eta_u L_f}{2} \left(1 - \langle \boldsymbol{u}_j^t, \boldsymbol{v}_i^t \rangle\right).$$

Using Lemma D.4 with $\boldsymbol{x}^t = \boldsymbol{u}_j^t$, $\boldsymbol{z}^t = \boldsymbol{v}_i^t$, and $\eta = \eta_u f(\boldsymbol{v}_i^t, \boldsymbol{u}_j^t)$, we get

$$\langle \boldsymbol{v}_i^t, \boldsymbol{u}_j^t - \boldsymbol{u}_j^{t+1} \rangle \leq -\frac{1}{\eta_u f(\boldsymbol{v}_i^t, \boldsymbol{u}_j^t)} \|\boldsymbol{u}_j^{t+1} - \boldsymbol{u}_j^t\|_2^2 \leq -\frac{1}{\eta_u} \|\boldsymbol{u}_j^{t+1} - \boldsymbol{u}_j^t\|_2^2.$$

Using Lemma D.4 with $\boldsymbol{x}^t = \boldsymbol{v}_i^t$, $\boldsymbol{z}^t = \frac{1}{|J|} \sum_{j \in J} \boldsymbol{u}_j^t$, and $\eta = \eta_c$, we get

$$\sum_{j \in J} \langle \boldsymbol{u}_j^t, \boldsymbol{v}_i^t - \boldsymbol{v}_i^{t+1} \rangle = |J| \langle \frac{1}{|J|} \sum_{j \in J} \boldsymbol{u}_j^t, \boldsymbol{v}_i^t - \boldsymbol{v}_i^{t+1} \rangle \leq -\frac{|J|}{\eta_c} \|\boldsymbol{v}_i^{t+1} - \boldsymbol{v}_i^t\|_2^2.$$

Using the above three inequalities, we can upper bound $\Phi^{t+1} - \Phi^t$:

$$\Phi^{t+1} - \Phi^t$$

$$= a_{t+1}\bigg(\frac{3}{4}\sum_{j\in J}\langle v_i^t, u_j^t - u_j^{t+1}\rangle + \frac{1}{4}\sum_{j\in J}\langle v_i^t, u_j^t - u_j^{t+1}\rangle$$

$$+ \sum_{j\in J}\langle u_j^t, v_i^t - v_i^{t+1}\rangle + \sum_{j\in J}\langle u_j^{t+1} - u_j^t, v_i^t - v_i^{t+1}\rangle\bigg) + (a_{t+1} - a_t)\sum_{j\in J}\big(1 - \langle u_j^t, v_i^t\rangle\big)$$

$$\leq a_{t+1}\bigg(-\frac{3}{4}\sum_{j\in J}\frac{\eta_u L_f}{2}\big(1 - \langle u_j^t, v_i^t\rangle\big) - \frac{1}{4}\sum_{j\in J}\frac{1}{\eta_u}\|u_j^{t+1} - u_j^t\|_2^2$$

$$- \frac{|J|}{\eta_c}\|v_i^{t+1} - v_i^t\|_2^2 + \sum_{j\in J}\|u_j^{t+1} - u_j^t\|_2 \cdot \|v_i^{t+1} - v_i^t\|_2\bigg) + (a_{t+1} - a_t)\sum_{j\in J}\big(1 - \langle u_j^t, v_i^t\rangle\big)$$

$$= a_{t+1}\bigg(-\frac{3\eta_u L_f}{8}\sum_{j\in J}\big(1 - \langle u_j^t, v_i^t\rangle\big)$$

$$- \sum_{j\in J}\bigg(\underbrace{\frac{1}{4\eta_u}\|u_j^{t+1} - u_j^t\|_2^2 + \frac{1}{\eta_c}\|v_i^{t+1} - v_i^t\|_2^2}_{\geq 2\sqrt{\frac{1}{4\eta_u\eta_c}\|u_j^{t+1}-u_j^t\|_2^2\|v_i^{t+1}-v_i^t\|_2^2}} - \|u_j^{t+1} - u_j^t\|_2 \cdot \|v_i^{t+1} - v_i^t\|_2\bigg)\bigg)$$

$$+ (a_{t+1} - a_t)\sum_{j\in J}\big(1 - \langle u_j^t, v_i^t\rangle\big)$$

$$\leq a_{t+1}\bigg(-\frac{3\eta_u L_f}{8}\sum_{j\in J}\big(1 - \langle u_j^t, v_i^t\rangle\big) - \sum_{j\in J}\Big(\underbrace{\sqrt{\tfrac{1}{\eta_u\eta_c}} - 1}_{\geq 0}\Big)\|u_j^{t+1} - u_j^t\|_2 \cdot \|v_i^{t+1} - v_i^t\|_2\bigg)$$

$$+ (a_{t+1} - a_t)\sum_{j\in J}\big(1 - \langle u_j^t, v_i^t\rangle\big)$$

$$\leq a_{t+1}\bigg(-\frac{3\eta_u L_f}{8}\sum_{j\in J}\big(1 - \langle u_j^t, v_i^t\rangle\big) + 0\bigg) + (a_{t+1} - a_t)\sum_{j\in J}\big(1 - \langle u_j^t, v_i^t\rangle\big)$$

$$= \bigg(\big(1 - \frac{3\eta_u L_f}{8}\big)a_{t+1} - a_t\bigg)\sum_{j\in J}\big(1 - \langle u_j^t, v_i^t\rangle\big)$$

$$= 0$$

where the last step is because $(1 - \frac{3\eta_u L_f}{8})a_{t+1} = a_t$.

We have shown that $\Phi^t$ is monotonically decreasing. Thus,

$$\frac{1}{2}\sum_{j\in J}\|u_j^T - v_i^T\|^2 = \frac{\Phi^T}{a_T} \leq \frac{\Phi^0}{a_T} \leq \frac{\sum_{j\in J}1}{a_T} = \big(1 - \frac{3\eta_u L_f}{8}\big)^T|J| \leq e^{-\frac{3\eta_u L_f}{8}T}|J| \leq \frac{1}{2}R^2$$

whenever $T \geq \frac{8}{3\eta_u L_f}\ln\frac{2|J|}{R^2}$. $\qquad\square$

**Corollary F.2** (of Lemma F.1). *Consider a system of $1$ creator $v_i^t$ and $|J|$ users $\{u_j^t\}_{j\in J}$, where the creator is recommended to all users at every time step. Assume:*

- *Initially, $\langle u_j^0, v_i^0\rangle \neq 0$ for every $j \in J$.*

- *There exists some constant $L_f > 0$ such that $|f(v_i, u_j)| \geq L_f > 0$ whenever $\langle v_i, u_j\rangle \neq 0$.*

- *$g(u_j, v_i) = \mathrm{sign}(\langle u_j, v_i\rangle)$.*

- *$\eta_c \leq \frac{\eta_u L_f}{2}$ and $0 \leq \eta_u < \frac{1}{2}$.*

*Then, for any $R > 0$, after at most $\frac{8}{3\eta_u L_f}\ln\frac{2|J|}{R^2}$ steps, the system will reach $R$-bi-polarization.*

*Proof.* Let $J^+ = \{j \in J : \langle \boldsymbol{u}_j^0, \boldsymbol{v}_i^0 \rangle > 0\}$ be the set of users with positive inner products with creator $i$ initially; let $J^- = \{j \in J : \langle \boldsymbol{u}_j^0, \boldsymbol{v}_i^0 \rangle < 0\}$. Let $\tilde{\boldsymbol{u}}_j^t = -\boldsymbol{u}_j^t$ for $j \in J^-$ and $\tilde{\boldsymbol{u}}_j^t = \boldsymbol{u}_j^t$ for $j \in J^+$. Then, the system consisting of $\{\tilde{\boldsymbol{u}}_j^t\}_{j \in J}$ and $\boldsymbol{v}_i^t$ satisfies the initial condition $\langle \tilde{\boldsymbol{u}}_j^0, \boldsymbol{v}_i^0 \rangle > 0$ in Lemma F.1. So, by Lemma F.1, it reaches $R$-consensus after at most $\frac{8}{3\eta_u L_f} \ln \frac{2|J|}{R^2}$ steps. Then by the reflection lemma (Lemma D.7), the original system, consisting of $\{\boldsymbol{u}_j^t\}_{j \in J}$ and $\boldsymbol{v}_i^t$, must reach $R$-bi-polarization. $\qquad \square$

### F.2 Inductive Step: Proof of Lemma 3.4

**Lemma F.3.** *Consider a system of $n \geq 1$ creators $\{\boldsymbol{v}_1^t, \ldots, \boldsymbol{v}_n^t\}$ and $|J|$ users $\{\boldsymbol{u}_j^t\}_{j \in J}$. Assume:*

- *Initially, $\langle \boldsymbol{v}_i^0, \boldsymbol{v}_{i'}^0 \rangle > 0$ for every $i, i'$, and $\langle \boldsymbol{v}_i^0, \boldsymbol{u}_j^0 \rangle > 0$ for every $i, j$.*

- *Assumptions of Lemma F.1.*

*Then, for any $R \in (0, 1)$, there exists a path of finite length that leads the initial state $(\boldsymbol{U}^0, \boldsymbol{V}^0)$ to $R$-consensus.*

*Proof.* Fix any $R \in (0, 1)$. Choose $R_1$ such that $\sqrt{(\frac{\eta_u}{\eta_c} + 2)4R_1} = R$. Clearly, $R_1 < R$. We construct a path that leads the state $(\boldsymbol{U}^0, \boldsymbol{V}^0)$ to $R$-consensus as follows.

*Step (1): Consider the subsystem of the first $n - 1$ creators and all users $J$. By induction, there exists a path of length $T_1 = L_{n-1,R_1} < +\infty$ that leads the subsystem to $(R_1, \boldsymbol{c}^{T_1})$-consensus with some $\boldsymbol{c}^{T_1} \in \mathbb{S}^{d-1}$. So, after these $T_1$ steps, all creators $i \in \{1, \ldots, n-1\}$ and all users $j \in J$ satisfy $\|\boldsymbol{v}_i^{T_1} - \boldsymbol{c}^{T_1}\| \leq R_1$ and $\|\boldsymbol{u}_j^{T_1} - \boldsymbol{c}^{T_1}\| \leq R_1$. Creator $n$ does not update during these $T_1$ steps, so $\boldsymbol{v}_n^{T_1} = \boldsymbol{v}_n^0$, and it still has positive inner products with the first $n - 1$ creators and all users by the convex cone property (Lemma D.2). Let's then consider the distance between creators $n$ and the consensus center $\boldsymbol{c}^{T_1}$: $\|\boldsymbol{v}_n^{T_1} - \boldsymbol{c}^{T_1}\|$. If $\|\boldsymbol{v}_n^{T_1} - \boldsymbol{c}^{T_1}\| \leq R$, then the system has satisfied $(R, \boldsymbol{c}^{T_1})$-consensus, so our construction is finished. Otherwise, $\|\boldsymbol{v}_n^{T_1} - \boldsymbol{c}^{T_1}\| > R$. We continue the construction as follows:*

*Step (2): Pick any user $j_0 \in J$, recommend creator $n$ to user $j_0$ for $T_2 = \frac{8}{3\eta_u L_f} \ln \frac{2}{R_1^2}$ steps, while recommending creator 1 to all other users.* From the $(R_1, \boldsymbol{c}^{T_1})$-consensus in step (1) we know $\|\boldsymbol{u}_{j_0}^{T_1} - \boldsymbol{c}^{T_1}\| \leq R_1$, so

$$\langle \boldsymbol{u}_{j_0}^{T_1}, \boldsymbol{c}^{T_1} \rangle = 1 - \tfrac{1}{2}\|\boldsymbol{u}_{j_0}^{T_1} - \boldsymbol{c}^{T_1}\|^2 \geq 1 - \tfrac{R_1^2}{2} > 1 - \tfrac{R^2}{2} \geq 1 - \tfrac{1}{2}\|\boldsymbol{v}_n^{T_1} - \boldsymbol{c}^{T_1}\|^2 = \langle \boldsymbol{v}_2^{T_1}, \boldsymbol{c}^{T_1} \rangle.$$

Thus, we can apply Lemma D.6 with $\boldsymbol{y} = \boldsymbol{c}^{T_1}$ to derive that, after these $T_2$ steps,

$$\langle \boldsymbol{v}_n^{T_1+T_2}, \boldsymbol{c}^{T_1} \rangle - \langle \boldsymbol{v}_n^{T_1}, \boldsymbol{c}^{T_1} \rangle \geq \tfrac{\eta_c}{\eta_u+\eta_c}\left( \langle \boldsymbol{u}_{j_0}^{T_1}, \boldsymbol{c}^{T_1} \rangle - \langle \boldsymbol{v}_n^{T_1}, \boldsymbol{c}^{T_1} \rangle - R_1 \right)$$

$$\geq \tfrac{\eta_c}{\eta_u+\eta_c}\left( 1 - \tfrac{R_1^2}{2} - \langle \boldsymbol{v}_n^{T_1}, \boldsymbol{c}^{T_1} \rangle - R_1 \right).$$

$$\implies \quad \langle \boldsymbol{v}_n^{T_1+T_2}, \boldsymbol{c}^{T_1} \rangle \geq \langle \boldsymbol{v}_n^{T_1}, \boldsymbol{c}^{T_1} \rangle + \tfrac{\eta_c}{\eta_u+\eta_c}\left( 1 - \tfrac{R_1^2}{2} - \langle \boldsymbol{v}_n^{T_1}, \boldsymbol{c}^{T_1} \rangle - R_1 \right). \quad (10)$$

For the inner product between creator $n$ and user $j_0$, by Lemma F.1 $\|\boldsymbol{v}_n^{T_1+T_2} - \boldsymbol{u}_{j_0}^{T_1+T_2}\| \leq R_1$, so

$$\langle \boldsymbol{v}_n^{T_1+T_2}, \boldsymbol{u}_{j_0}^{T_1+T_2} \rangle = 1 - \tfrac{1}{2}\|\boldsymbol{v}_n^{T_1+T_2} - \boldsymbol{u}_{j_0}^{T_1+T_2}\|^2 \geq 1 - \tfrac{R_1^2}{2}. \quad (11)$$

Consider the inner products between creator $n$ and the first $n - 1$ creators and the users in $J \setminus \{j_0\}$. Because the first $n-1$ creators and the users in $J \setminus \{j_0\}$ form $(R_1, \boldsymbol{c}^{T_1})$-consensus at time step $T_1$, by Observation 3.2, they still form $(R_1, \boldsymbol{c}^{T_1})$-consensus at time step $T_1 + T_2$, so $\|\boldsymbol{v}_i^{T_1+T_2} - \boldsymbol{c}^{T_1}\| \leq R_1$ and $\|\boldsymbol{u}_j^{T_1+T_2} - \boldsymbol{c}^{T_1}\| \leq R_1$. This implies, for $i \neq n$,

$$\langle \boldsymbol{v}_n^{T_1+T_2}, \boldsymbol{v}_i^{T_1+T_2} \rangle \geq \langle \boldsymbol{v}_n^{T_1+T_2}, \boldsymbol{c}^{T_1} \rangle - \|\boldsymbol{v}_i^{T_1+T_2} - \boldsymbol{c}^{T_1}\|$$

$$\geq \langle \boldsymbol{v}_2^{T_1+T_2}, \boldsymbol{c}^{T_1} \rangle - R_1$$

$$(10) \geq \langle \boldsymbol{v}_n^{T_1}, \boldsymbol{c}^{T_1} \rangle + \tfrac{\eta_c}{\eta_u+\eta_c}\left( 1 - \tfrac{R_1^2}{2} - \langle \boldsymbol{v}_n^{T_1}, \boldsymbol{c}^{T_1} \rangle - R_1 \right) - R_1, \quad (12)$$

and for $j \in J \setminus \{j_0\}$,

$$
\begin{aligned}
\langle v_n^{T_1+T_2}, u_j^{T_1+T_2} \rangle &\geq \langle v_n^{T_1+T_2}, c^{T_1} \rangle - \| u_j^{T_1+T_2} - c^{T_1} \| \\
&\geq \langle v_2^{T_1+T_2}, c^{T_1} \rangle - R_1 \\
(10) \quad &\geq \langle v_n^{T_1}, c^{T_1} \rangle + \tfrac{\eta_c}{\eta_u+\eta_c} \left( 1 - \tfrac{R_1^2}{2} - \langle v_n^{T_1}, c^{T_1} \rangle - R_1 \right) - R_1. \qquad (13)
\end{aligned}
$$

*Step (3): Consider the subsystem of the first $n - 1$ creators and all users $J$. By induction, there exists a path of length $T_3 = L_{n-1,R_1} < +\infty$ that leads the subsystem to $(R_1, c^{T_1+T_2+T_3})$-consensus with some $c^{T_1+T_2+T_3} \in \mathbb{S}^{d-1}$. So, we have $\| v_i^{T_1+T_2+T_3} - c^{T_1+T_2+T_3} \| \leq R_1$ for every $i \in \{1, \dots, n-1\}$ and $\| u_j^{T_1+T_2+T_3} - c^{T_1+T_2+T_3} \| \leq R_1$ for every $j \in J$, and $v_n^{T_1+T_2+T_3} = v_n^{T_1+T_2}$. Consider the inner product between creator $n$ and any of the first $n - 1$ creators $i \in \{1, \dots, n-1\}$. By the convex cone property (Lemma D.2),*

$$
\begin{aligned}
\langle v_n^{T_1+T_2+T_3}, v_i^{T_1+T_2+T_3} \rangle &= \langle v_n^{T_1+T_2}, v_i^{T_1+T_2+T_3} \rangle \\
\text{Lemma D.2} \quad &\geq \min \left\{ \langle v_n^{T_1+T_2}, v_i^{T_1+T_2} \rangle, \min_{j \in J} \langle v_n^{T_1+T_2}, u_j^{T_1+T_2} \rangle \right\} \\
\text{by (11), (12), (13)} \quad &\geq \min \left\{ \langle v_n^{T_1}, c^{T_1} \rangle + \tfrac{\eta_c}{\eta_u+\eta_c} \left( 1 - \tfrac{R_1^2}{2} - \langle v_n^{T_1}, c^{T_1} \rangle - R_1 \right) - R_1, \ 1 - \tfrac{R_1^2}{2} \right\} \\
&= \langle v_n^{T_1}, c^{T_1} \rangle + \tfrac{\eta_c}{\eta_u+\eta_c} \left( 1 - \tfrac{R_1^2}{2} - \langle v_n^{T_1}, c^{T_1} \rangle - R_1 \right) - R_1 \qquad (14)
\end{aligned}
$$

where the last equality is because, under the assumption of $\| v_n^{T_1} - c^{T_1} \| > R = \sqrt{\left( \tfrac{\eta_u}{\eta_c} + 2 \right) 4 R_1}$,

$$
\begin{aligned}
&\langle v_n^{T_1}, c^{T_1} \rangle + \tfrac{\eta_c}{\eta_u+\eta_c} \left( 1 - \tfrac{R_1^2}{2} - \langle v_n^{T_1}, c^{T_1} \rangle - R_1 \right) - R_1 \\
&= \tfrac{\eta_u}{\eta_u+\eta_c} \langle v_n^{T_1}, c^{T_1} \rangle + \tfrac{\eta_c}{\eta_u+\eta_c} \left( 1 - \tfrac{R_1^2}{2} - R_1 \right) - R_1 \\
&= \tfrac{\eta_u}{\eta_u+\eta_c} \left( 1 - \tfrac{1}{2} \| v_2^{T_1} - c^{T_1} \|^2 \right) + \tfrac{\eta_c}{\eta_u+\eta_c} \left( 1 - \tfrac{R_1^2}{2} - R_1 \right) - R_1 \\
&\leq \tfrac{\eta_u}{\eta_u+\eta_c} \left( 1 - \tfrac{1}{2} \left( \tfrac{\eta_u}{\eta_c} + 2 \right) 4 R_1 \right) + \tfrac{\eta_c}{\eta_u+\eta_c} \left( 1 - \tfrac{R_1^2}{2} - R_1 \right) - R_1 \\
&\leq \max \left\{ 1 - \tfrac{1}{2} \left( \tfrac{\eta_u}{\eta_c} + 2 \right) 4 R_1, \ 1 - \tfrac{R_1^2}{2} - R_1 \right\} - R_1 \\
&= 1 - \tfrac{R_1^2}{2} - R_1 - R_1 \ \leq \ 1 - \tfrac{R_1^2}{2}.
\end{aligned}
$$

From (14) and $\| v_i^{T_1+T_2+T_3} - c^{T_1+T_2+T_3} \| \leq R_1$,

$$
\begin{aligned}
\langle v_n^{T_1+T_2+T_3}, c^{T_1+T_2+T_3} \rangle &\geq \langle v_n^{T_1+T_2}, v_i^{T_1+T_2+T_3} \rangle - \| v_i^{T_1+T_2+T_3} - c^{T_1+T_2+T_3} \| \\
&\geq \langle v_n^{T_1}, c^{T_1} \rangle + \tfrac{\eta_c}{\eta_u+\eta_c} \left( 1 - \tfrac{R_1^2}{2} - \langle v_n^{T_1}, c^{T_1} \rangle - R_1 \right) - 2R_1.
\end{aligned}
$$

Using 1 to minus the above inequality, we obtain

$$
1 - \langle v_n^{T_1+T_2+T_3}, c^{T_1+T_2+T_3} \rangle \ \leq \ \tfrac{\eta_u}{\eta_u+\eta_c} \left( 1 - \langle v_n^{T_1}, c^{T_1} \rangle \right) + \tfrac{\eta_c}{\eta_u+\eta_c} \left( \tfrac{R_1^2}{2} + R_1 \right) + 2R_1.
$$

Let $F^t = 1 - \langle v_n^t, c^t \rangle$, then

$$
F^{T_1+T_2+T_3} \ \leq \ \tfrac{\eta_u}{\eta_u+\eta_c} F^{T_1} + \tfrac{\eta_c}{\eta_u+\eta_c} \left( \tfrac{R_1^2}{2} + R_1 \right) + 2R_1. \qquad (15)
$$

*Repeat steps (2) and (3) for $K$ times.* Then, using (15) for $K$ times,

$$
\begin{aligned}
F^{T_1+K(T_2+T_3)} &\leq \tfrac{\eta_u}{\eta_u+\eta_c} F^{T_1+(K-1)(T_2+T_3)} + \tfrac{\eta_c}{\eta_u+\eta_c}\left(\tfrac{R_1^2}{2}+R_1\right) + 2R_1 \\[4pt]
&\leq \tfrac{\eta_u}{\eta_u+\eta_c}\left(\tfrac{\eta_u}{\eta_u+\eta_c} F^{T_1+(K-2)(T_2+T_3)} + \tfrac{\eta_c}{\eta_u+\eta_c}\left(\tfrac{R_1^2}{2}+R_1\right) + 2R_1\right) + \tfrac{\eta_c}{\eta_u+\eta_c}\left(\tfrac{R_1^2}{2}+R_1\right) + 2R_1 \\[2pt]
&\;\;\vdots \\[2pt]
&\leq \left(\tfrac{\eta_u}{\eta_u+\eta_c}\right)^K F^{T_1} + \left(1 + \tfrac{\eta_u}{\eta_t+\eta_c} + \cdots + \left(\tfrac{\eta_u}{\eta_t+\eta_c}\right)^{K-1}\right)\left(\tfrac{\eta_c}{\eta_u+\eta_c}\left(\tfrac{R_1^2}{2}+R_1\right) + 2R_1\right) \\[4pt]
&\leq \left(\tfrac{\eta_u}{\eta_u+\eta_c}\right)^K \cdot 1 + \tfrac{1}{1-\frac{\eta_u}{\eta_u+\eta_c}}\left(\tfrac{\eta_c}{\eta_u+\eta_c}\left(\tfrac{R_1^2}{2}+R_1\right) + 2R_1\right) \\[4pt]
&= \left(\tfrac{\eta_u}{\eta_u+\eta_c}\right)^K + \tfrac{R_1^2}{2} + R_1 + \tfrac{\eta_u+\eta_c}{\eta_c}2R_1 \\[4pt]
&\leq \tfrac{R_1^2}{2} + \tfrac{R_1^2}{2} + R_1 + \tfrac{\eta_u+\eta_c}{\eta_c}2R_1 \;\leq\; \left(\tfrac{\eta_u}{\eta_c}+2\right)2R_1,
\end{aligned}
$$

by choosing $K = \frac{\ln\frac{2}{R_1^2}}{\ln\frac{\eta_u+\eta_c}{\eta_u}} \leq \frac{\eta_u+\eta_c}{\eta_c}\ln\frac{2}{R_1^2}$. This means that, after repeating steps (2) and (3) for $K$ times, we must have

$$
\begin{aligned}
\|\boldsymbol{v}_n^{T_1+K(T_2+T_3)} - \boldsymbol{c}^{T_1+K(T_2+T_3)}\| &= \sqrt{2\big(1 - \langle \boldsymbol{v}_n^{T_1+K(T_2+T_3)}, \boldsymbol{c}^{T_1+K(T_2+T_3)}\rangle\big)} \\[4pt]
&= \sqrt{2F^{T_1+K(T_2+T_3)}} \;\leq\; \sqrt{2\big(\tfrac{\eta_u}{\eta_c}+2\big)2R_1} \;=\; R.
\end{aligned}
$$

The above inequality, together with the fact that other creators $i \neq n$ and all users in $J$ already satisfy $(R_1 \leq R, \boldsymbol{c}^{T_1+K(T_2+T_3)})$-consensus after step (3), implies that the whole system has reached $(R, \boldsymbol{c}^{T_1+K(T_2+T_3)})$-consensus.

The length of the path constructed above is at most:

$$
T_1 + K(T_2+T_3) \;\leq\; L_{n-1,R_1} + \tfrac{\eta_u+\eta_c}{\eta_c}\ln\tfrac{2}{R_1^2}\left(\tfrac{8}{3\eta_u L_f}\ln\tfrac{2|J|}{R_1^2} + L_{n-1,R_1}\right) \;=\; L_{n,R} \;<\; +\infty,
$$

which is finite. $\qquad\square$

**Lemma F.4.** *Consider a subsystem of $n$ creators $\{\boldsymbol{v}_1^t, \ldots, \boldsymbol{v}_n^t\}$ and $|J|$ users $\{\boldsymbol{u}_j^t\}_{j\in J}$. Assume:*

- *Initially, the first $n-1$ creators and all users are in $R_0$-consensus: $\|\boldsymbol{v}_i^0 - \boldsymbol{c}\| \leq R_0$, $\|\boldsymbol{u}_j^0 - \boldsymbol{c}\| \leq R_0$, with $0 < R_0 < \frac{\eta_c}{5(\eta_c+\eta_u)}$.*

- $\langle \boldsymbol{v}_n^0, \boldsymbol{u}_{j_0}^0\rangle > 0$ *for some $j_0 \in J$.*

- $g(\boldsymbol{u}_j, \boldsymbol{v}_i) = \text{sign}(\langle \boldsymbol{u}_j, \boldsymbol{v}_i\rangle)$.

- *Assumption of Lemma F.1.*

*Then, for any $R \in (0,1)$, there exists a path of finite length that leads the initial state $(\boldsymbol{U}^0, \boldsymbol{V}^0)$ to $R$-consensus.*

*Proof.* First, we recommend creator $n$ to user $j_0$ for $T = \frac{8}{3\eta_u L_f}\ln\frac{2}{R_0^2}$ steps, while recommending other creators to other users arbitrarily. Applying Lemma D.6 with $\boldsymbol{y} = \boldsymbol{u}_{j_0}^0$, we get

$$
\langle \boldsymbol{v}_n^T, \boldsymbol{u}_{j_0}^0\rangle - \langle \boldsymbol{v}_n^0, \boldsymbol{u}_{j_0}^0\rangle \;\geq\; \tfrac{\eta_c}{\eta_u+\eta_c}\left(\langle \boldsymbol{u}_{j_0}^0, \boldsymbol{u}_{j_0}^0\rangle - \langle \boldsymbol{v}_n^0, \boldsymbol{u}_{j_0}^0\rangle - R_0\right) \;=\; \tfrac{\eta_c}{\eta_u+\eta_c}\left(1 - \langle \boldsymbol{v}_n^0, \boldsymbol{u}_{j_0}^0\rangle - R_0\right). \tag{16}
$$

On the other hand, because the first $n-1$ creators and all users in $J \setminus \{j_0\}$ form an $(R_0, \boldsymbol{c})$-consensus at time step 0, according to Observation 3.2, they still form an $(R_0, \boldsymbol{c})$-consensus at time step $T$, so $\|\boldsymbol{v}_i^T - \boldsymbol{c}\| \leq R_0$ for every $i \in \{1, \ldots, n-1\}$. This implies, for every $i \in \{1, \ldots, n-1\}$,

$$
\langle \boldsymbol{v}_n^T, \boldsymbol{v}_i^T\rangle - \langle \boldsymbol{v}_n^T, \boldsymbol{u}_{j_0}^0\rangle \;\geq\; -\|\boldsymbol{v}_i^T - \boldsymbol{u}_{j_0}^0\| \;\geq\; -\|\boldsymbol{v}_i^T - \boldsymbol{c}\| - \|\boldsymbol{c} - \boldsymbol{u}_{j_0}^0\| \;\geq\; -2R_0. \tag{17}
$$

Adding (16) and (17) and moving $\langle v_n^0, u_{j_0}^0 \rangle$ to the right side, we get

$$\langle v_n^T, v_i^T \rangle \geq \langle v_n^0, u_{j_0}^0 \rangle + \tfrac{\eta_c}{\eta_u+\eta_c}\left(1 - \langle v_n^0, u_{j_0}^0 \rangle - R_0\right) - 2R_0$$

$$= \tfrac{\eta_u}{\eta_u+\eta_c}\langle v_n^0, u_{j_0}^0 \rangle + \tfrac{\eta_c}{\eta_u+\eta_c}\left(1 - R_0\right) - 2R_0$$

$$> 0 + \tfrac{\eta_c}{\eta_u+\eta_c}\left(1 - R_0\right) - 2R_0 > 0,$$

under the condition of $R_0 < \frac{\eta_c}{5(\eta_u+\eta_c)}$. Moreover, for every $j \in J \setminus \{j_0\}$, because $\|u_j^T - v_i^T\| \leq \|u_j^T - c\| + \|c - v_i^T\| \leq 2R_0$,

$$\langle v_n^T, u_j^T \rangle \geq \langle v_n^T, v_i^T \rangle - \|u_j^T - v_i^T\| \geq \tfrac{\eta_c}{\eta_u+\eta_c}\left(1 - R_0\right) - 4R_0 > 0.$$

For $j_0$, by Lemma F.1, $\|v_n^T - u_{j_0}^T\| \leq R_0$, so

$$\langle v_n^T, u_{j_0}^T \rangle = 1 - \tfrac{1}{2}\|v_n^T - u_{j_0}^T\|^2 \geq 1 - \tfrac{R_0^2}{2} > 0.$$

For the inner product between any creator $i \in \{1, \ldots, n-1\}$ and the users:

$$\langle v_i^T, u_{j_0}^T \rangle \geq \langle v_i^T, v_n^T \rangle - \|v_n^T - u_{j_0}^T\| \geq \tfrac{\eta_c}{\eta_u+\eta_c}\left(1 - R_0\right) - 2R_0 - R_0 = \tfrac{\eta_c}{\eta_u+\eta_c}\left(1 - R_0\right) - 3R_0 > 0;$$

$$\forall j \in J \setminus \{j_0\}, \quad \langle v_i^T, u_j^T \rangle = 1 - \tfrac{1}{2}\|v_i^T - u_j^T\|^2 \geq 1 - \tfrac{1}{2}\left(\|v_i^T - c\| + \|c - u_j^T\|\right)^2 > 1 - \tfrac{1}{2}(2R_0)^2 > 0.$$

All of the "> 0" inequalities above show that the system of $\{v_i^T\}_{i \in [n]}$ and $\{u_j^T\}_{j \in J}$ satisfies the condition of Lemma F.3. So, there exists a path of finite length $T_2 < +\infty$ that leads the system to $R$-consensus by Lemma F.3. The total length of path $T + T_2 = \frac{8}{3\eta_u L_f} \ln \frac{2}{R_0^2} + T_2 < +\infty$ is finite. □

**Lemma 3.4.** *Suppose $\eta_c \leq \frac{\eta_u L_f}{2}$ and $\eta_u < \frac{1}{2}$. For any $R > 0$, for almost every state $(U^t, V^t)$ in the state space, there exists a path $(U^t, V^t) \to (U^{t+1}, V^{t+1}) \to \cdots \to (U^{t+T}, V^{t+T})$ of finite length that leads to an $R$-bi-polarization state $(U^{t+T}, V^{t+T})$.*

*Proof.* We prove this lemma by induction on the number of creators $n$. The case for $n = 1$ directly follows from Corollary F.2 which shows that, for any system of $n = 1$ creator and $|J|$ users with no $\langle v_i^0, u_j^0 \rangle = 0$, there exists a path of length at most $L_1^R = \frac{8}{3\eta_u L_F} \ln \frac{2|J|}{R^2} < +\infty$ that leads to $R$-bi-polarization.

Consider $n \geq 2$. Consider the subsystem consisting of the first $n - 1$ creators $\{v_1^t, \ldots, v_{n-1}^t\}$ and all users. Let $R_0 = \frac{\eta_c}{6(\eta_c+\eta_u)}$. By induction, there exists a path of finite length $T_1 = L_{n-1}^{R_0} < +\infty$ that leads the subsystem to $R_0$-bi-polarization, with some vector $c_0 \in \mathbb{S}^{d-1}$, so every $v_i^{T_1}$ is $R_0$-close to $+c_0$ or $-c_0$, for $i \neq n$, and every $u_j^{T_1}$ is $R_0$-close to $+c_0$ or $-c_0$. Define:

$$\tilde{v}_i^t = \begin{cases} v_i^t & \text{if } v_i^{T_1} \text{ is } R_0\text{-close to } +c \\ -v_i^t & \text{if } v_i^{T_1} \text{ is } R_0\text{-close to } -c \end{cases} \forall i \neq n, \qquad \tilde{u}_j^t = \begin{cases} u_j^t & \text{if } u_j^{T_1} \text{ is } R_0\text{-close to } +c \\ -u_j^t & \text{if } u_j^{T_1} \text{ is } R_0\text{-close to } -c \end{cases} \forall j \in J.$$

By definition, we have

$$\|\tilde{v}_i^{T_1} - c_0\| \leq R_0, \quad \forall i \neq n, \qquad \|\tilde{u}_j^{T_1} - c_0\| \leq R_0, \quad \forall j \in J.$$

This means that $\{\tilde{v}_i^{T_1}\}_{i \neq n}$ and $\{\tilde{u}_j^{T_1}\}_{j \in J}$ form an $(R_0, c_0)$-consensus. Consider creator $n$. Let

$$\tilde{v}_n^t = \begin{cases} v_n^t & \text{if } \langle v_n^{T_1}, \tilde{u}_{j_0}^{T_1} \rangle > 0 \text{ for some } j_0 \in J \\ -v_n^t & \text{if } \langle v_n^{T_1}, \tilde{u}_j^{T_1} \rangle < 0 \text{ for all } j \in J. \end{cases}$$

(The case where $\langle v_n^{T_1}, \tilde{u}_j^{T_1} \rangle = 0$ for some $j \in J$ is ignored because the initial states that can lead to such states have measure 0.) By definition, we have

$$\langle \tilde{v}_n^{T_1}, \tilde{u}_{j_0}^{T_1} \rangle > 0 \text{ for some } j_0 \in J.$$

Note that, at time step $T_1$, the system consisting of $\{\tilde{v}_i^{T_1}\}_{i \in [n]}$ and $\{\tilde{u}_j^{T_1}\}_{j \in J}$ satisfies the condition of Lemma F.4, so there exists a path of length $T_2 = \tilde{L}_n^R < +\infty$ that leads the system to $R$-consensus. Then by the reflection lemma (Lemma D.7), the original system $\{v_i^t\}_{i \in [n]}$, $\{u_j^t\}_{j \in J}$ must reach $R$-bi-polarization. The total length of path that leads to this $R$-bi-polarization is $L_n^R = T_1 + T_2 = L_{n-1}^{R_0} + \tilde{L}_n^R < +\infty$. □

# G   Missing Proofs from Section 4

## G.1   Proof of Proposition 4.2

Let $R > 0$ be any small number. Let $\boldsymbol{c}_1, \ldots, \boldsymbol{c}_{\lfloor n/k \rfloor} \in \mathbb{R}^d$ be $\lfloor n/k \rfloor$ vectors that satisfy $B(\boldsymbol{c}_\ell, 2R) \cap B(\boldsymbol{c}_{\ell'}, 2R) = \emptyset$ for $\ell \neq \ell'$, where $B(\boldsymbol{c}, R)$ is the ball centered at $\boldsymbol{c}$ with radius $R$: $\{\boldsymbol{x} \in \mathbb{R}^d : \|\boldsymbol{x} - \boldsymbol{c}\|_2 \leq R\}$. Consider user and creator features $(\boldsymbol{U}^t, \boldsymbol{V}^t)$ that satisfy: every ball $B(\boldsymbol{c}_\ell, R)$ ($\ell = 1, \ldots, \lfloor n/k \rfloor$) contains $k$ creator vectors, and every user vector $\boldsymbol{u}_j^t$ is in one of the balls $B(\boldsymbol{c}_\ell, R)$. By definition, $(\boldsymbol{U}^t, \boldsymbol{V}^t)$ form $\lfloor n/k \rfloor$ clusters. We show that, after one step of update, the new state $(\boldsymbol{U}^{t+1}, \boldsymbol{V}^{t+1})$ must still form $\lfloor n/k \rfloor$ clusters. Consider any user $j$. Suppose $\boldsymbol{u}_j^t \in B(\boldsymbol{c}_\ell, R)$, then the distance from $\boldsymbol{u}_j^t$ to any creator $\boldsymbol{v}_i^t \in B(\boldsymbol{c}_\ell, R)$ is at most $2R$:

$$\|\boldsymbol{u}_j^t - \boldsymbol{v}_i^t\| \leq 2R.$$

The distance from $\boldsymbol{u}_j^t$ to any creator $\boldsymbol{v}_{i'}^t$ not in $B(\boldsymbol{c}_\ell, R)$ is greater than $2R$:

$$\|\boldsymbol{u}_j^t - \boldsymbol{v}_{i'}^t\| > 2R$$

because $\boldsymbol{v}_{i'}^t$ is in some other ball $B(\boldsymbol{c}_{\ell'}, R)$ that satisfies $B(\boldsymbol{c}_{\ell'}, 2R) \cap B(\boldsymbol{c}_\ell, 2R) = \emptyset$. This implies that the inner products between user $j$ and the creators in ball $B(\boldsymbol{c}_\ell, R)$ are greater than that with the creators in other ball:

$$\forall \boldsymbol{v}_i^t \in B(\boldsymbol{c}_\ell, R), \ \ \langle \boldsymbol{u}_j^t, \boldsymbol{v}_i^t \rangle = 1 - \frac{1}{2}\|\boldsymbol{u}_j^t - \boldsymbol{v}_i^t\|_2^2 \geq 1 - \frac{1}{2}(2R)^2 > 1 - \frac{1}{2}\|\boldsymbol{u}_j^t - \boldsymbol{v}_{i'}^t\| = \langle \boldsymbol{u}_j^t, \boldsymbol{v}_i^t \rangle, \ \ \forall \boldsymbol{v}_{i'}^t \in B(\boldsymbol{c}_{\ell'}, R).$$

Since $B(\boldsymbol{c}_\ell, R)$ contains $k$ creators, these $k$ creators are the $k$-most relevant ones to user $j$, so user $j$ will only be recommended these creators. Then, by applying Observation 3.2 to each of the $\lfloor n/k \rfloor$ balls separately, we see that each ball is a $R$-consensus and hence absorbing. So, the new state $(\boldsymbol{U}^{t+1}, \boldsymbol{V}^{t+1})$ still forms $\lfloor n/k \rfloor$ clusters with these $\lfloor n/k \rfloor$ balls.

## G.2   Proof of Proposition 4.3

The $d$-dimensional simplex centered at the original has $d+1$ vectors with negative inner products with each other. They form $d+1$ clusters. Since user-creator pairs with negative inner product $\langle \boldsymbol{u}_i, \boldsymbol{v}_j \rangle < 0$ are not recommended, recommendations only happen within each cluster. By Observation 3.2, each cluster is absorbing, so the whole system is stable, keep forming $d+1$ clusters forever.

# H   Additional Discussion on Real-World Recommender Systems

Here we further discuss real-world recommender systems' properties and designs that are currently not covered in our main paper. We plan to generalize our model in the future to further capture these features and discuss insightful findings, but having them in the current paper may be a distraction to our main findings.

## H.1   User and Creator Retention and Activeness

In our current model, the users and creators will stay in the system from the start to the end. However, in real-world recommender systems, users and creators may leave the platform either permanently or for a certain period. Meanwhile, new users and creators will join the platform. Such join and leave dynamics are also influenced by the recommendations' relevance and diversity, which further complicate the problem. Moreover, users and creators have different activeness levels on the platform, e.g., some users may watch a lot more videos than others, and some creators may post a lot more creations, these effects will also be strongly correlated with the dual influence of the recommender system.

## H.2   Creation Quality

Creation quality is a major factor influencing users' feedback in addition to the creation style, e.g., well-made cuisine videos could also be fun and liked by gamers and pet lovers, which we need more than a collaborative filtering type of modeling like our current model to capture such features. A

potential solution to boost both long-term system diversity and single-shot recommendation diversity is to design mechanisms that can incentivize creators to create higher-quality videos instead of changing their creation styles.

## H.3 Cold Start

Cold Start is widely used in real-world recommender systems for newly published items. Due to the lack of user-item interactions on new items, the systems randomly recommend these new items to users and collect data for collaborative filtering. In our current model, if we consider the creators creating new items in each time step under their current time creation style, then cold start guarantees the conditions in Theorem 3.3. But if we consider the system to have good enough content understanding ability and can accurately predict the new creations' embeddings, the cold start is not necessary and our model and results in the top-$k$ truncation and threshold truncation parts are valid. We also highlight a subtle difference between cold start and random traffic, if cold start is used on creators instead of items, then after the creator is exposed to users a certain number of times, the system will not guarantee to provide a non-zero probability of recommending this creator, and thus the conditions in Theorem 3.3 may not hold.

