# OpenReview forum: "User-Creator Feature Polarization in Recommender Systems with Dual Influence"
_NeurIPS.cc/2024/Conference — NeurIPS 2024 poster_

### Official Review · Reviewer_JScD · 2024-06-29

**Soundness:** 4
**Presentation:** 2
**Contribution:** 4
**Rating:** 7
**Confidence:** 5

**Summary:**

This paper models dynamics of both users and creators in a recommender system. The user features shift in the direction of the content recommended to them. The creator dynamics are strategically motivated i.e. they try to align content to attract their audience, to increase profit.

The authors then provide sufficient conditions for this model of dual dynamics to converge to polarization under a natural assumption that each creator has some non zero probability of being recommended to every user.

The paper then discusses four real world recommendation designs, and whether they cause polarization or multiple clusters etc. They also provide results on synthetic and Movielens data complementing theory results and show that certain recommender designs do lead to polarization vs diverse clusters.

**Strengths:**

- This paper is the first to consider dynamics of both users and creators in a recommender systems and provides sufficient analytic conditions for polarization
- They apply this theory to 4 natural designs: (1) Top-k,  (2) Truncation, (3)Diversity boosting and (4) Lower bounding probability. They show that rules (3, 4) lead to polarization and rule (1) leads to diverse clusters. This section is particularly insightful.
- The experimental evaluation with synthetic and Movielens data is also insightful and complements the theory. The softmax probability leads to diminishing creator and recommendation diversity over time. They also study top-k probability and show how lower k is better for higher creator diversity, recommendation relevance.

**Weaknesses:**

A criticism I had while reading the paper are gaps in literature for the discussion on dynamics in recommender systems. In addition to [Eilat & Rosenfeld] referenced in the introduction, [1,2,3,4,5,6] consider creator dynamics in recommender systems. These works assume static user features and provide results on content at equilibrium and user welfare. In the context of these works, it would be beneficial to highlight how your work is the first to consider both creator and user dynamics.

[1] A Game-Theoretic Approach to Recommendation Systems with Strategic Content Providers (Ben-Porat and Tennenholtz)

[2] Supply-side equilibria in recommender systems (Jagadeesan et al)

[3] How Bad is Top-k Recommendation under Competing Content Creators? (Yao et al)

[4] Modeling content creator incentives on algorithm-curated platforms (Hron et al)

[5] Producers Equilibria and Dynamics in Engagement-Driven Recommender Systems (Acharya et al)

[6] User Welfare Optimization in Recommender Systems with Competing Content Creators (Yao et al)

**Questions:**

-  I understand the motivation for the form of user update in equation (3) . In each recommendation step an item is recommended and the user preference shifts in that direction, this is like in [Dean and Morgenstern]. Can you motivate the update in Eq (4), is this myopically optimal for the creator to do, and how does it generalize [Eilat & Rosenfeld]?
- Minor Typo? For Figure 6,  larger $\rho$ seems to lead to higher creator diversity (green curve).

**Limitations:**

The authors discuss limitations of their results in the paper.

---

> ### Author Rebuttal · Authors · 2024-08-07
>
> > Q1: Can you motivate the update in Eq (4)? Is this myopically optimal for the creator to do, and how does it generalize [Eilat & Rosenfeld]?
>
> [[Eilat & Rosenfeld]](https://arxiv.org/pdf/2302.04336) assumes that creators aim to maximize exposure (defined as the sum of inner products between user embeddings and the creator embedding) minus the content adjustment cost (see their Section 2.1 and equation (6)). They show that such creators will move towards the average of the user embeddings by some step size (their equation (11)). In our notations, their update is $v_i^{t+1} = \mathcal P(v_i^t + \frac{\eta_c}{|J_i^t|} \sum_{j\in J_i^t} u_j^t)$, which is the special case of our equation (4) where $g$ is the constant positive function $g(u_j, v_i) = 1$.
>
> Nevertheless, as we wrote in Lines 122 - 129, we are motivated by a different type of creators: rating-maximizing creators. This means that the $g(u_j, v_i)$ function should have the same sign as $\langle u_j, v_i\rangle$. Intuitively, for all the users $J_i^t$ being recommended creator $i$, if the user likes the item, $\langle u_j^t, v_i^t\rangle > 0 \implies g(u_j^t, v_i^t) > 0$, then the creator is incentivized to move towards that user to receive more positive rating from that user. Otherwise, $\langle u_j^t, v_i^t\rangle < 0 \implies g(u_j^t, v_i^t) < 0$, the creator is incentivized to move away from the user in order to be recommended less often to that user (to avoid being negatively-rated by the user). Taking both scenarios into account, the creator moves toward the weighted average $\sum_{j\in J_i^t} g(u_j^t, v_i^t) u_j^t$, which gives equation (4). Under this particular assumption on $g$, our equation (4) does not capture [[Eilat & Rosenfeld]](https://arxiv.org/pdf/2302.04336).
>
>
>
> > Q2: Minor Typo? For Figure 6, larger $\rho$ seems to lead to higher creator diversity (green curve).
>
> Yes, this is a minor typo. The figure shows that a higher $\rho$ leads to higher creator diversity but also higher tendency to polarization at the same time.  A possible explanation for this phenomenon is (similar to our reasoning in Lines 307 to 309): the system polarizes into two balanced clusters which actually have a large average pairwise distance.  So in this case, Tendency to Polarization may be a better measure for diversity loss than the Creator Diversity measure (average pairwise distance).

---

> > ### Comment · Reviewer_JScD · 2024-08-07
> > **Acknowledging rebuttal**
> >
> > Thanks for the addressing my questions, I stand with my assessment of accepting this paper.

---

### Official Review · Reviewer_Gmcw · 2024-07-14

**Soundness:** 3
**Presentation:** 3
**Contribution:** 3
**Rating:** 7
**Confidence:** 3

**Summary:**

The paper explores the dynamics between users and content creators in recommender systems, highlighting the dual influence where users’ preferences are shaped by recommendations and creators modify their content to align with what is more likely to be recommended. The study defines a model called user-creator feature dynamics to capture these interactions, demonstrating that such systems are prone to polarization, resulting in a loss of diversity. The paper then examines various approaches to mitigate polarization and improve diversity, finding that relevancy-optimizing methods, such as top-k recommendations, can prevent polarization more effectively than traditional diversity-promoting approaches.

**Strengths:**

The paper provides an interesting perspective by addressing the mutual influence between users and creators in recommender systems. The theoretical results and experimental validation using both synthetic and real-world data look credible. The writing is overall easy to follow.

**Weaknesses:**

1. There are two lines of works focusing on modeling content creator dynamics and user preference evolving dynamics that are neglected by the authors. I listed several representative works and it would be great to include a comprehensive literature review regarding these works in the related work section.

2. One of your main observation (larger $\beta$ leads to higher creator diversity and alleviated polarization) is actually pointed out in [1] under a similar model, where content creators compete for a fixed user population (see section 3.2 in [1]). And another main observation in section 5.3 that smaller $k$ improves diversity does not echo the result in [2], which shows that larger $k$ improves the total creator utilities. It would be better to include some detailed discussions regarding these two works.

3. The user/creator preference updating dynamics need more justifications and empirical evidence.

4. The dynamical model makes some sense to me, but it would be more interesting to understand whether the observations still hold in the presence of noise. If the noisy version is hard to analyze theoretically, additional simulation results could also be valuable.


[1]. Modeling Content Creator Incentives on Algorithm-Curated Platforms
[2]. How Bad is Top-K Recommendation under Competing Content Creators?
[3]. Online recommendations for agents with discounted adaptive preferences
[4]. Recommender systems as dynamical systems: Interactions with viewers and creators
[5]. Learning from a learning user for optimal recommendations
[6]. Supply-side equilibria in recommender systems

**Questions:**

1. In theorem 3.3, how does the convergence rate depending on the temperature parameter $\beta$? I ask this because when $\beta\rightarrow +\infty$, the softmax recommendation strategy is equivalent to the top-1 recommendation strategy. In this case, Proposition 4.2 predicts that the top-1 recommendation lead to $n$ clusters rather than bi-polarization, which seems to contradict theorem 3.3.
2. I do not fully get why the specific forms of function $f$ and $g$ do not affect the analysis. Is it because your main results only depend on the range of $f$ and $g$?
3. In the experiments, the range of $\beta$ is quite conservative. I'm curious about the result under a larger range of $\beta$.

**Limitations:**

yes

---

> ### Author Rebuttal · Authors · 2024-08-07
>
> > Weakness 1: related works.
>
> Thank you for listing those related works! We will discuss them in the revision. We also provide comparisons between some of those works and our work in a table in our global response.
>
>
> > Weakness 2: It would be better to include some detailed discussions regarding these two works [1] [2].
> >
> > [1] Modeling Content Creator Incentives on Algorithm-Curated Platforms
> >
> > [2] How Bad is Top-K Recommendation under Competing Content Creators?
>
> [1] indeed also observes that a larger $\beta$ leads to higher creator diversity. We view our work as complementary to [1]. Our findings corroborate those of [1] despite two key differences between our setting and that of [1]: in [1], creators maximize exposure while we consider creators who maximize user engagement; second, [1]'s user population is fixed while ours is adaptive. This helps us understand that there may be some fundamental mitigation strategies (such as selecting $\beta$), which have similar effects regardless of changes to the underlying problem formulation.
>
> [2] shows that a larger $k$ improves the social welfare, defined to be the total utility/relevance of the users (which is also the total utility of creators in their model), assuming a fixed user population. In contrast, we focus on the diversity/polarization of creators, and we have adaptive users. So the $k$ plays a different role in our setting: a larger $k$ leads to worse creator diversity.
>
> > Weakness 3: the user/creator preference updating dynamics need more justifications and empirical evidence.
>
> Our user preference update model is a generalization of [[Dean & Morgenstern, EC 22]](https://arxiv.org/pdf/2205.13026), which models user update as
> $u^{t+1}_j = \mathcal{P}(u^t_j + \eta \langle v_i^t, u_j^t\rangle v_i^t)$.
> Our work replaces the inner produce with a general function $f(v_i^t, u_j^t)$ (our Equation 3), with some constraints outlined on lines 113 - 121 of our paper. The motivation for this update rule, as outlined by [[Dean & Morgenstern, EC 22]](https://arxiv.org/pdf/2205.13026), is the "biased assimilation" phenomenon, which is in turn inspired by the opinion polarization literature.  (We mentioned this in Lines 107, 115, and 241.)
>
> Our creator update model, as we wrote in Lines 122 - 129, is motivated by rating-maximizing creators. This means that the $g(u_j, v_i)$ function has the same sign as $\langle u_j, v_i\rangle$. Intuitively, for all the users $J_i^t$ being recommended creator $i$, if the user likes the item, $\langle u_j^t, v_i^t\rangle > 0 \implies g(u_j^t, v_i^t) > 0$, then the creator is incentivized to move towards that user to receive more positive rating from that user. Otherwise, $\langle u_j^t, v_i^t\rangle < 0 \implies g(u_j^t, v_i^t) < 0$, the creator is incentivized to move away from the user in order to be recommended less often to that user (to avoid being negatively-rated by the user). Taking both scenarios into account, the creator moves toward the weighted average $\sum_{j\in J_i^t} g(u_j^t, v_i^t) u_j^t$, which gives our update rule (4).
>
> If any additional details would help improve the justification of the user/creator update rules, please let us know. This is an important aspect of our work and we would like to polish it as much as possible.
>
>
> > Weakness 4: it would be more interesting to understand whether the observations still hold in the presence of noise.
>
> In the newly uploaded PDF, we provide simulation results where the user and creator updates include normally distributed unbiased noises inside the projection operator:
> $u_j^{t+1} = P (u_j^t + \eta_u f( v_{i_j^t}^t, u_j^t) v_{i_j^t}^t + \eta_u \epsilon_j^t)$ where $\epsilon_j^t \sim Normal(0, \sigma^2I)$
> and
> $v_i^{t+1} = P ( v_i^t + \frac{\eta_c}{|J_i^t|} \sum_{j \in J_i^t} g(u_j^t, v_i^t) u_j^t + \eta_c \epsilon_i^t)$ where $\epsilon_i^t \sim Normal(0, \sigma^2 I)$.
>
> We observe that a small noise still leads to near polarization, while a large noise (large $\sigma$) reduces the tendency to polarization. And the observation that top-k recommendation reduces polarization still holds. (see Figures R2 - R5 in the uploaded PDF.)
>
> > Q1: how does the convergence rate depend on the temperature parameter $\beta$? I ask this because when $\beta\rightarrow +\infty$, the softmax recommendation strategy is equivalent to the top-1 recommendation strategy, which by Proposition 4.2 leads to $n$ clusters rather than bi-polarization, which seems to contradict theorem 3.3.
>
> Finite $\beta$ and infinite $\beta$ are qualitatively different. Finite $\beta$ leads to bi-polarization, while $\beta = +\infty$ is equivalent to top-1 recommendation and does not lead to bi-polarization.  Indeed, the rate of convergence to bi-polarization with a large but finite $\beta$ might be slow (see the "Tendency to Polarization" plot in Figure R1 in the PDF we uploaded during rebuttal).  Our Theorem 3.3 is more of an asymptotic result.  Analyzing the convergence rate would be an interesting yet challenging direction for future work.
>
> > Q2: I do not fully get why the specific forms of function do not affect the analysis.
>
> For $f$, our results and analysis only require the assumptions in Lines 114 - 121 that $f(v_i, u_j)$ has the same sign as $\langle v_i, u_j\rangle$ and is two-sided bounded $L_f \le |f(v_i, u_j)| \le 1$.
>
> For $g$, our current analysis assumes the specific form of $g(u_j, v_i) = sign(\langle u_j, v_i\rangle)$. We believe our analysis (in particular, Lemma E.1) can be generalized to other $g$ functions satisfying similar assumptions as $f$ (but as of now this generalization remains an open problem).
>
>
> > Q3: results under a large range of $\beta$.
>
> In Figure R1 of the uploaded PDF, we provide experiment results for $\beta \in [0, 10]$ and $+\infty$. We note that $\beta = 10$, although not very large, has similar effects as $\beta=+\infty$ (top-1 recommendation) in 1000 time steps, because the softmax probability of non-max creators when $\beta =10$ is very close to 0.

---

> > ### Comment · Reviewer_Gmcw · 2024-08-13
> > **Re: Rebuttal**
> >
> > I thank the authors for their detailed response, which addressed most of my concerns. And I appreciate the summarized table which emphasizes the contribution. I decide to maintain my score, leaning towards acceptance.

---

### Official Review · Reviewer_EsTG · 2024-07-14

**Soundness:** 2
**Presentation:** 3
**Contribution:** 3
**Rating:** 6
**Confidence:** 4

**Summary:**

This paper studies how recommendations become polarized over the long run when user and creator features dynamically change over time. The authors theoretically prove that, under the assumption that every creator can be recommended to every user with some non-zero probability, recommender systems will eventually converge to polarization. They also simulate some real-world models, including top-k recommendation, truncation, diversity boosting, and lower-bounding probabilities in a long-term setting. The key observation is that top-k recommendation (i.e., only recommending top-k items to users) can reduce polarization to some extent, while existing diversity-boosting methods will worsen polarization when user/creator features dynamically change over time in the system.

**Strengths:**

1. The authors provide both theoretical and empirical evidence showing that relevance-focused recommendations (as opposed to diversity-focused recommendations), which harm diversity in a static setting, are actually effective in improving diversity in the long term. This observation is somewhat counter-intuitive to previous beliefs, making it very interesting.
2. The authors conducted simulations with both synthetic data and real-world data (i.e., Movielens) using four diversity and relevance-related measures. Additionally, the analysis with sensitivity parameters in softmax is insightful and supports the authors' main claim.
3. Studying diversity in a dynamic setting is novel.

**Weaknesses:**

1. Despite the novelty and interestingness, I have concerns about the key assumptions of the theoretical and empirical analyses. The assumption that all items can be recommended to users is not realistic. In practice, almost all recommender systems rely on top-k recommendations for either effectiveness or resource constraints like screen size. For example, on platforms like Netflix or Amazon, customers can only see a certain number of items on the webpage (i.e., p=0 for items that users can't see). Even if they can scroll down and the system continually recommends new items, they cannot physically see all items in the system. Thus, I believe the top-k setting is the most realistic and natural for real-world scenarios, and this seems like a hole in the authors' analyses. In this sense, the measures for empirical analysis should also only consider top-k items, not all items.
2. For the real-world designs, it would be more extensive if users included trustworthiness-aware recommender systems that consider dynamic/continual settings. For example, [1] consider performance difference between two different user groups when the user/item features are continually updated over time in the systems.
3. For the analysis with Movielens, considering the interaction timestamp in the simulation would more accurately reflect real-world scenarios, for example, for determining the true labels.

[1] Yoo et al., Ensuring User-side Fairness in Dynamic Recommender Systems, WWW'24

**Questions:**

1. Please address the points I raised in Weaknesses.
2. (Minor) Are both consensus and bi-polarization conceptually polarization?
3. (Minor) How are the initial user/creator embeddings initialized in the Movielens experiment?

**Limitations:**

The authors adequately addressed the limitations.

---

> ### Author Rebuttal · Authors · 2024-08-07
>
> > Q1: Please address the points I raised in Weaknesses.
>
> > W1: The assumption that all items can be recommended to users is not realistic. ... Customers can only see a certain number of items on the webpage (i.e., p=0 for items that users can't see).
>
>
> First, we note that customers not seeing some items due to screen size limit does not mean that those items are recommended with probability p=0. The set of items seen by a customer is randomly drawn from a distribution over items which can have positive probability on all items, even if only $k$ items are shown.
>
>
> Second, we argue that non-zero probability of recommendation is realistic, even in large-scale real-world recommendation systems used by Yahoo [[Marlin et al, 2009]](https://dl.acm.org/doi/10.1145/1639714.1639717), [[Li et al, 2010]](https://dl.acm.org/doi/10.1145/1772690.1772758), and Kuaishou [[Gao et al, 2022]](https://arxiv.org/pdf/2208.08696.pdf). As noted in our Section 4.4, practical recommendation systems insert small random traffic (uniformly random recommendations, or missing-at-random MAR [[Yang et al, 2018]](https://dl.acm.org/doi/10.1145/3240323.3240355) data) to improve recommendation diversity [[Section 2.2 of Moller et al, 2018]](https://www.tandfonline.com/doi/full/10.1080/1369118X.2018.1444076) or to explore users' interests for unseen contents [[Gao et al, 2022]](https://arxiv.org/pdf/2208.08696.pdf) or for debias purposes [[Liu et al, 2023]](https://dl.acm.org/doi/10.1145/3582002). These interventions will cause all recommendation probabilities to be non-zero (although they may be very small).
>
> > ... In this sense, the measures for empirical analysis should also only consider top-k items, not all items.
>
> Note that our measures RD and RR include the recommendation probability $p_{ij}$ (where $p_{ij} = 0$ for non-top-k creators), so RD and RR do consider top-k items. Our other two measures, CD and TP, aim to measure the diversity of the entire creator pool, independent of the recommendation scheme, so they do not consider the recommendation probability for users.
>
>
> > W2: For the real-world designs, it would be more extensive if users included trustworthiness-aware recommender systems that consider dynamic/continual settings. For example, [1] consider performance difference between two different user groups when the user/item features are continually updated over time in the systems.
>
> We thank the reviewer for the suggestion and we agree that fairness-aware recommendation systems are important in practice. However, we want to make the best use of the limited space in the main article to introduce the notion of dual influence and outline its relationship with polarization in recommendation systems. Whether a system polarizes its users or creators is by itself an interesting trustworthiness question, separated from fairness considerations. Moreover, [1] does not study strategic content creators in recommendation systems and has fundamental differences with our work. Applying this work in case of adaptive users and creators will require a careful re-designing of the method proposed in [1]. We see the consideration of fairness-aware systems as a deeply important area, and hope that our work on dual influence will inspire future research to consider both fairness-aware recommendation and dual influence (as we fully agree with you that most realistic settings would consider both aspects). We are happy to add further discussions on fairness-aware recommendation systems and how our work relate to them in the appendix in the final version.
>
> > W3: For the analysis with Movielens, considering the interaction timestamp in the simulation would more accurately reflect real-world scenarios, for example, for determining the true labels.
>
> We agree with the reviewer that this will be an interesting simulation. However, we need to model the dual dynamics (that include both user updates and creator updates) in the system, such dynamics are not currently captured in any publicly available dataset that we are aware of. Since we need to model dual dynamics, solely relying on existing MovieLens data (or any other dataset) is not enough, we need to create pseudo-real-world data which simulates these dual dynamics.
>
>
> > Q2: (Minor) Are both consensus and bi-polarization conceptually polarization?
>
> Yes.
>
> > Q3: (Minor) How are the initial user/creator embeddings initialized in the Movielens experiment?
>
> The initialization is done by fitting a two-tower model [Huang et al.] on the existing MovieLens ratings data and we use the tower tops as the initial user and creator embeddings.
>
> ----
> [Huang et al.] Learning Deep Structured Semantic Models for Web Search using Clickthrough Data.  CIKM, 2013.

---

> ### Comment · Reviewer_EsTG · 2024-08-07
>
> Thanks for the detailed response. However, I still feel that my question about top-k recommendation has not been adequately answered. Let me clarify the question further.
>
> 1. In the paper, lines 192-193 state, "In particular, we consider the top-k recommendation policy where each user is recommended only he k most relevant creators, so pt ij = 0 if i is not one of k creators i′ that maximize ⟨vt i′ , ut j ⟩.", meaning that p=0 in the case of top-K recommendation. My thought was that nearly all recommendation scenarios are essentially top-K recommendation practices, which means p=0 in these cases as well. Could you provide examples where this is not the case?
> As mentioned in line 188-189, involves filtering out items unlikely to be relevant to a user and then recommending from the remaining items. This filtering process is typically based on relevance scores, where only the highest scores are retained. In this sense, it seems equivalent to "only K items are shown." Therefore, I am unclear on why there would be a non-zero probability when only K items are shown. What is the difference between top-K recommendation in Section 4-1 and these real-world scenarios?
>
> 2. Regarding random interventions, are you suggesting or is it possible that these interventions occur on top of top-k recommendations (i.e., so if there are n interventions, there would be K+n items in the list)? I believe this scenario could indeed have a non-zero probability even within the top-k recommendation practice. But without this scenario, I think most recommendation scenarios are basically top-k recommendations in Section 4-1. What are your thoughts on this?

---

> > ### Author Response · Authors · 2024-08-09
> > **Clarifications on "top-k recommendation", "non-zero probability of not shown items", and "random intervention"**
> >
> > We would like to provide some clarification regarding item delivery. When referring to "top-$k$”, we actually mean “top-$k$ truncation". For a given user $u_i$, we compute probability $p_{ij}$ for each creator $v_j$. Let $p^{(k)}$ be the $k^{\text{th}}$ largest $p_{ij}$, then all $p_{ij} < p^{(k)}$ are set to 0. Creators are then recommended to $u_i$ based on the remaining nonzero probability $p_{ij}$. This filtering process corresponds to the (first) recall stage commonly found in large-scale recommendation systems with two (or more) stages (see, e.g., [[Youtube's DNN recommendation paper]](https://static.googleusercontent.com/media/research.google.com/en//pubs/archive/45530.pdf)).
> >
> > Top-$k$ truncation does not mean showing $k$ items to the users. Let's say that a user sees $L$ items in a short time frame. In real-world scenarios, the value of $L$ is usually smaller than $k$ and can depend on factors like screen size. Moreover, the set of $L$ items shown to the user is a random sample from a distribution of items. Reviewer EsTG wrote "(p=0 for items that users can't see)" and "Therefore, I am unclear on why there would be a non-zero probability when only K items are shown". That seems to be a misunderstanding. Even if an item $j$ is not shown to the user in one time step, its probability $p_{ij}$ can still be $>0$, and this item can be sampled and shown to the user in the next time step.
> >
> > The random traffic intervention (which has been used in large-scale recommendation systems as we discussed in Lines 225 - 227; see also [[KuaiRand, page 4, left column (ii)]](https://arxiv.org/pdf/2208.08696)) bypasses all the steps (scoring, filter, etc) in the multi-stage recommendation system and replaces the candidates returned by the multi-stage recommendation process with randomly chosen candidates at a low probability. The random traffic intervention does not cause efficiency loss for the system.  And with random traffic, a user might be recommended any possible creator (namely, all $p_{ij} > 0$), even if $L$ is small.

---

> > > ### Comment · Reviewer_EsTG · 2024-08-09
> > >
> > > Thank you for the reply. I agree the screen size L could be much smaller than K.
> > > Let me ask you something more just to ensure my understanding on this setting is correct. Whether we use top-K truncation or use all N creators, we are assuming that recommendations are based on their probability distribution, correct? Rather than just recommending the Top-X creators with the highest probabilities.

---

> ### Author Response · Authors · 2024-08-10
> **Correct, recommendations are based on probability distribution.**
>
> Yes, we are recommending creators based on their probability distribution.
>
> Without top-k truncation, each creator $j$ (among the N total creators) is recommended with $p_{ij}$ probability.
>
> With top-k truncation, the N creators with the $k$ highest relevance scores pass the filter/truncation stage, and these $k$ creators are then recommended with probability proportional to $p_{ij}$ (calculated and normalized using these top $k$ relevance scores). We are not deterministically selecting the top-$L$ ($L < k$) creators to show to the user.

---

> ### Comment · Reviewer_EsTG · 2024-08-11
>
> Thank you for the clarifications. Most of my concerns have been resolved, so I'll increase the credits. I hope the authors can include their rebuttal in the final version.

---

> > ### Author Response · Authors · 2024-08-12
> >
> > Thank you!  We will surely include our rebuttal in the final version.

---

### Author Rebuttal · Authors · 2024-08-07

We thank the reviewers for the helpful comments, especially the provided related works. Here, we provide a table to compare our work with those works (and some works that were already cited in our paper). We will add this table to an additional related work section in our appendix. We want to highlight that, while previous works have studied dynamic creators and dynamic users separately, "our work is the first to consider both creator and user dynamics", as pointed out by reviewer JScD.

| Works            | Adaptive Users | Adaptive Creators | Creator Reward  | Characterizing Dynamics or Equilibrium Behavior | Creation Change Model |
| ---------------- | ---------|-----  | -------------- | --------------------- | ------------------- |
| Our Work         | Yes | Yes | User Engagement | Dynamics | Conditioned on previous time step; implicit cost of content adjustment|
| Eilat & Rosenfeld [1] | No | Yes | Exposure        | Dynamics | Conditioned on previous time step; explicit cost of content adjustment |
| Yao et al [2]      | No  | Yes | User Engagement | Dynamics | Freely choose without cost |
| Jagadeesan et al [3] | No  | Yes | Exposure        | Equilibrium  | Freely choose with cost    |
| Hron and Krauth et al [4] | No | Yes |  Exposure    | Equilibrium  | Freely choose without cost |
| Ben-Porat and Tennenholtz [5] | No | Yes |  Exposure    | Equilibrium  | Freely choose without cost |
| Acharya et al [6] | No | Yes |  User Engagement    | Equilibrium  | Freely choose without cost |
| Yao et al [7] | No | Yes |  Reward designed by a social-welfare-maximizing platform    | Dynamics  | Freely choose without cost |
| Dean and Morgenstern [8] | Yes | No* |  N/A    | Dynamics  | N/A |
| Yao et al [9] | Yes | No* |  N/A    | Dynamics  | N/A |
| Agarwal and Brown [10] | Adaptive and adversarial | No* |  N/A    | Dynamics  | N/A |

*: These works study the design of recommendation algorithms for the platform with a fixed set of items, without explicitly modeling the content creators.

---

[1] Eilat and Nir Rosenfeld. Performative Recommendation: Diversifying Content via Strategic Incentives. ICML 2023.

[2] Yao et al. How Bad is Top-K Recommendation under Competing Conten Creators? ICML 2023.

[3] Jagadeesan et al. Supply-Side Equilibria in Recommender Systens. NeurIPS 2023.

[4] Hron and Krauth et al. Modeling Content Creator Incentives on Algorithm-Curated Platforms. ICLR 2023.

[5] Ben-Porat and Tennenholtz. A Game-Theoretic Approach to Recommendation Systems with Strategic Content Providers. NeurIPS 2018.

[6] Acharya et al. Producers Equilibria and Dynamics in Engagement-Driven Recommender Systems. ArXiv 2024.

[7] Yao et al. User Welfare Optimization with Competing Ccontent Creators. ArXiv 2024.

[8] Dean and Morgenstern. Preference Dynamics Under Personalized Recommendations. EC, 2022.

[9] Yao et al. Learning from a Learning User for Optimal Recommendations. ICML 2022.

[10] Agarwal and Brown. Online recommendations for agents with discounted adaptive preferences. ALT 2024.

---

### Decision · Program_Chairs · 2024-09-25

**Decision:**

Accept (poster)

**Comment:**

The paper studies the dual influence of recommender systems on users and creators, and provides theoretical proof on the convergence towards polarization under the model and certain conditions, and presented findings showing relevance focusing systems such as top-K recommendations rather than diversity promoting systems are more effective at preventing polarization.  Reviewers find the work solid and the result interesting. Some minor comments are missing references and clarification on assumptions were addressed during rebuttal.